

# The relative importance of wind and hydroclimate drivers in modulating wind-blown dust emissions in Earth system models

Xinzhu Li[1], Longlei Li[2], Yan Feng[3], and Xin Xi[1]

[1]Department of Geological and Mining Engineering and Sciences, Michigan Technological University, Houghton, MI, USA
[2]Department of Earth and Atmospheric Sciences, Cornell University, Ithaca, NY, USA
[3]Environmental Science Division, Argonne National Laboratory, Lemont, IL, USA

**Correspondence:** Longlei Li (ll859@cornell.edu) and Xin Xi (xinxi@mtu.edu)

**Abstract.**

Windblown dust emissions are subject to large uncertainties in Earth system models (ESMs), yet model discrepancies in dust variability and its physical drivers remain poorly understood. This study evaluates the consistency of 21 ESMs in simulating the climatological distribution and interannual variability of global dust emissions and applies dominance analysis to quantify the
relative influence of near-surface wind speed and five hydroclimate variables (precipitation, soil moisture, specific humidity, air temperature, leaf area index) across different climate zones. In hyperarid regions, the models exhibit poor agreement in dust variability, with only 10% of pairwise comparisons showing significant positive correlations. Most models capture the dominant wind control except GFDL-ESM4 which display dominant hydroclimate influence (wind contributing 42%) and high spatial variability. In arid and semiarid regions, dust variability is shaped by a dual effect of land surface memory: models
with consistent hydroclimate variability converge in dust responses, while those with divergent hydroclimate representations show increased disagreement. While all models capture the expected increase of hydroclimate influence with decreasing aridity, the extent of this transition varies by model, resulting in greater model disagreement regarding the relative importance of wind and hydroclimate drivers in arid/semiarid regions. Implementing the Kok et al. (2014) scheme in CESM reduces the wind contribution from 86% to 64% in hyperarid regions and from 56% to 46% in arid regions, indicating enhanced hydroclimate
influence compared to the Zender et al. (2003) scheme. These findings underscore the importance of improving hydroclimate and land surface representations for reducing uncertainties in dust emission responses to climate variability and change.

## 1 Introduction

Windblown dust from global drylands is an essential element of the Earth's biogeochemical cycle, but has become a global concern due to its transboundary, multifaceted impacts on the climate, public health, agriculture, and socioeconomic well-
being. Dust emission is modulated by a number of atmospheric and land surface parameters that can be grouped into three broad drivers: sediment supply, sediment availability, and wind erosivity, which collectively determine the timing, location, duration, and intensity of dust events (Xi, 2023). The most abundant *sediment supply* is typically found in low relief regions with thick layers of fine, unconsolidated materials produced by weathering, fluvial, and/or aeolian processes (Bryant, 2013). These fine sediments, however, are not always available for dust entrainment due to the inhibition by soil moisture and surface



armoring, such as vegetation, soil crusts and non-erodible coarse particles (Bullard et al., 2011). The *sediment availability* is modulated primarily by hydroclimate conditions and land use practices (e.g., desertification, afforestation), which affect the minimum or *threshold wind velocity* required to initiate mobilization. *Wind erosivity* is dominated by high wind events which produce sufficient drag to mobilize dust particles through sandblasting and saltation. Depending on the relative importance of the three drivers, dust emission may fall into one of three distinct regimes: *supply-limited*, where a lack of suitable-sized

sediments restricts dust emission; *availability-limited*, where fine sediments are present but are protected against erosion; and *transport capacity-limited*, where sediments are dry and exposed but the near-surface winds are too weak to trigger dust entrainment.

The physical drivers have been incorporated into coupled aerosol-climate models and Earth system models (ESMs) to represent the environmental control of the dust cycle. In many models, sediment supply is represented by a time-invariant

dust source function which approximates the abundance of erodible materials on a scale of 0 to 1 (e.g., Ginoux et al., 2001; Zender et al., 2003). High values are generally found in topographic depressions which contain thick layers of alluvial or lacustrine deposits (Prospero et al., 2002). Over erodible areas, models generally assume an unlimited sediment supply without accounting for sediment depletion or replenishment over time (Zhang et al., 2016a). The hydroclimate modulation of sediment availability is represented by multiple parameters or processes. A common approach involves scaling vertical dust fluxes by the

bare soil fraction to account for the presence of non-erodible surfaces such as water, snow, and vegetation. Most models also incorporate the effect of soil moisture on the enhancement of threshold wind velocity (Fécan et al., 1999). In addition, a drag partition term is commonly used to reduce the wind stress acting on erodible surfaces in the presence of vegetation and surface roughness elements (Marticorena and Bergametti, 1995; Shao et al., 2011). To account for the dependence on wind erosivity, horizontal dust fluxes are computed as the third or fourth power of wind speeds above the threshold velocity. Because of this

nonlinear relationship and the skewed distribution of wind speeds, global dust emissions are dominated by rare, high wind events (Cowie et al., 2015; Bergametti et al., 2017). Accurate model representations of these wind events remains challenging since peak-wind generation mechanisms, such as downdrafts from moist convection, often occur at spatial scales smaller than the typical grid spacing of global models (Cakmur et al., 2004; Grini et al., 2005; Ridley et al., 2013; Zhang et al., 2016b).

The Aerosol Comparisons between Observations and Models (AeroCom) initiative and Coupled Model Intercomparison

Project (CMIP) have facilitated the comparison of global dust cycle in ESMs (Textor et al., 2006; Huneeus et al., 2011; Kim et al., 2014; Wu et al., 2020; Gliß et al., 2021; Zhao et al., 2022; Kim et al., 2024). In general, the contemporary dust column burden is reasonably constrained by ground- and satellite-based aerosol optical depth (AOD) observations near dust source regions, resulting in better model agreement than those in dust emission and deposition. Knippertz and Todd (2012) argued that model tunings to match satellite observations—often through dust source functions—can induce a compensational effect

between dust emission and deposition, both of which lack reliable observational constraints at global scales. As a result, dust emissions exhibit substantial model discrepancies, as well as large biases in reproducing the observed dust variability (Huneeus et al., 2011; Wu et al., 2020; Gliß et al., 2021; Zhao et al., 2022). The model discrepancies and biases can, at least partly, be explained by the different dust emission schemes implemented in ESMs. Early-generation schemes use predefined, time-invariant dust source functions to represent the spatially varying soil erodibility. They may also use simplified parameterizations



without explicitly accounting for the effect of sandblasting efficiency on dust vertical fluxes. In contrast, newer schemes use more mechanistic representations of the dust flux dependence on land surface conditions and sandblasting efficiency, thereby reducing or eliminating the need for dust source functions. While such schemes offer improved model physics, they require more extensive input parameters, some of which are either not well constrained by observations or are not properly represented in models (Marticorena and Bergametti, 1995; Shao et al., 1996; Lu and Shao, 1999; Kok et al., 2014b). One example is

the choice of wind speed in dust flux calculations. Some models use 10-m winds for simplicity, whereas others use friction velocity, which more accurately capture the wind stress exerted on soil surfaces. However, estimating friction velocity requires specifying the surface roughness length, a parameter that lacks robust global observation constraints.

For ESMs using the same dust scheme, substantial discrepancies can also result from different model resolutions, tuning parameters, and coupled physical parameterizations. For example, the bare soil fraction is computed based on the land/water

mask, land cover, green vegetation fraction and snow cover, all of which may vary by model. In particular, vegetation fraction may be prescribed from a fixed climatology or computed interactively with dynamic vegetation simulations. Additional differences arise from the soil column representations (e.g., number of layers, layer thickness), soil thermal and hydraulic properties, and formulations of runoff, evaporation and infiltration processes. Together, these processes affect the topsoil water content and ultimately the threshold wind velocity. Furthermore, parameterizations of planetary boundary layer processes and subgrid-

scale wind variability strongly affect the model capability in simulating the high wind events pertinent to dust emissions. Given the strong coupling between dust emission and multiple atmospheric and land surface processes, it is not surprising that dust emission fluxes are strongly model-dependent.

In this study, we draw an analogy from Koster et al. (2009)'s interpretation of root-zone soil moisture and treat the dust emission flux as a model-specific quantity, characterized by a dynamic range determined by the dust emission scheme and

the coupled processes within each ESM. The dust emission fluxes produced by various ESMs essentially represent different approximations of the true state which they are trying to reproduce but has no direct observations to validate against. As such, these fluxes are expected to differ in climatological means and variability characteristics. The true value of dust emission model simulations thus lies not necessarily in their absolute magnitudes, but rather in their spatiotemporal variability and sensitivity to the physical drivers. If the variability of physical drivers is consistent between models, one would expect some degree of

correlation in the simulated dust fluxes. For example, stronger winds combined with drier soils should lead to more emissions, regardless of which model is used. To date, the extent of inter-model agreement in the dust variability and sensitivity to physical drivers within ESMs remains poorly understood. In particular, the relative importance of wind erosivity and hydroclimate-modulated sediment erodibility within individual ESMs has not been evaluated. By quantifying the collective and relative contributions of wind and hydroclimate drivers in explaining the dust variability simulated by a range of ESMs and reanalysis

products, this study provides new insights into model discrepancies and biases in global dust emission simulations.

The remainder of this paper is structured as follows. Section 2 describes the ESMs and aerosol reanalysis datasets considered in this study, as well as the dominance analysis technique for quantifying the relative importance of individual physical drivers of dust emissions. Section 3 presents the intercomparison of the climatological mean, geographic contribution, and interannual



**Table 1.** Summary of dust emission parameterizations in the Earth system models and aerosol reanalysis datasets. DSF, dust source function. $u_*$, friction velocity. $u$, 10-m wind speed. $\omega$, soil moisture. $r_0$, surface roughness. ERS, European Remotes Sensing satellite.

| Model | $D_{max}$ | Wind | DSF | Emission Threshold | Reference |
|---|---|---|---|---|---|
| CESM2-CAM-Zender | 10 | $u_*^3$ | Zender et al. (2003), Truncated | Fécan et al. (1999); Marticorena and Bergametti (1995); Constant $z_0$ of 100 $\mu$m | Albani et al. (2015) |
| CESM2-WACCM-Zender | Same as CESM2-CAM-Zender | | | | |
| CESM2-CAM-Kok | 10 | $u_*^3$ | No DSF | Kok et al. (2014b) scheme; Limited $z_0$ effect | Li et al. (2022a) |
| E3SM2-Zender | Same as CESM2-CAM-Zender except using the original DSF. | | | | Feng et al. (2022) |
| E3SM3-Kok | Same as CESM2-CAM-Kok | | | | Xie et al. (2025) |
| CanESM5-1 | Bulk | $u_*^3$ | Tegen et al. (2002) | Peng et al. (2012); Monthly $z_0$ from ERS | Sigmond et al. (2023) |
| CNRM-ESM2.1 | 20 | $u_*^3$ | No DSF | Fécan et al. (1999); Marticorena and Bergametti (1995) | Nabat et al. (2015) |
| EC-Earth3-AerChem | 20 | $u_*^3$ | Tegen et al. (2002) | No $\omega$ effect; Marticorena and Bergametti (1995); Monthly $z_0$ from ERS | Van Noije et al. (2021) |
| GISS-E2.1-OMA | 32 | $u^3$ | Ginoux et al. (2001) | Shao et al. (1996); Emission permitted for ERS $z_0$<0.1 cm only. | Miller et al. (2006) |
| GISS-E2.1-MATRIX | Same as GISS-E2.1-OMA | | | | |
| GISS-E2.2-OMA | Same as GISS-E2.1-OMA | | | | |
| GFDL-ESM4 | 20 | $u_*^3$ | Ginoux et al. (2001) | No emission if above a $\omega$ threshold; Fixed $u_{*th}$ depending on land types | Evans et al. (2016) |
| HadGEM3-GC31 | 63 | $u_*^3$ | Ginoux et al. (2001) | Fécan et al. (1999); No $r_0$ effect | Woodward (2011) |
| UKESM1.0 | 63 | $u_*^3$ | No DSF | Fécan et al. (1999): No $r_0$ effect | Woodward et al. (2022) |
| INM-CM5.0 | Bulk | $u_*^4$ | No DSF | No emission if $\omega$ >10 kg/$m^2$; No $r_0$ effect | Volodin and Kostrykin (2016) |
| IPSL-CM6A-LR | Bulk | $u^3$ | Schulz et al. (2009) | Balkanski et al. (2004); Fixed $u_{th}$ depending on soil types and slopes | Szopa et al. (2013) |
| MRI-ESM2.0 | 20 | $u_*^3$ | No DSF | Shao et al. (1996); Shao and Lu (2000) | Yukimoto et al. (2019) |
| MIROC6 | 10 | $u^3$ | No DSF | Takemura et al. (2009); Fixed $u_{th}$ of 6.5 m/s. | Tatebe et al. (2019) |
| MIROC-ES2L | Same as MIROC6 | | | | Hajima et al. (2020) |
| MPI-ESM-1.2 | Bulk | $u_*^3$ | Tegen et al. (2002) | Fécan et al. (1999); Cheng et al. (2008); Monthly $z_0$ from ERS | Tegen et al. (2019) |
| NorESM2 | Same as CESM2-CAM-Zender | | | | Seland et al. (2020) |
| MERRA2 | 20 | $u^3$ | Ginoux et al. (2001) | Marticorena and Bergametti (1995) | Randles et al. (2017) |
| JRAero | Same as MRI-ESM2.0 | | | | Yumimoto et al. (2017) |

variability of dust emissions, and the relative contributions of wind and hydroclimate drivers to the simulated dust variability
within each ESM. The conclusions are summarized in Section 4.





## 2 Data and Approach

### 2.1 ESMs and reanalysis products

Table 1 summarizes the dust emission parameterizations used in the ESMs and reanalysis products considered in this study. These include fully-coupled simulations from 18 CMIP6 models for the period of 1950–2014. Unless otherwise specified,
we use the first ensemble member (r1i1p1f1) from each model. In the CMIP6 archive, two configurations of the Community Earth System Model (CESM) share the same dust scheme of Zender et al. (2003) (hereafter referred to as the Zender scheme), but use different atmospheric modules: Community Atmosphere Model (CAM) vs. Whole Atmosphere Community Climate Model (WACCM). The key difference between CAM and WACCM lies in their vertical extent and representation of upper atmospheric processes. The three GISS-E2 models use the same dust scheme described in Miller et al. (2006), but differ by
model version (2.1 vs. 2.2) and aerosol microphysics schemes: One-Moment Aerosol (OMA; ensemble member r1i1p3f1) vs. Multiconfiguration Aerosol TRacker of mIXing state (MATRIX; ensemble member r1i1p5f1) (Miller et al., 2021). UKESM1.0 is built upon the HadGEM3-GC3.1 model, which share the same dust scheme of Woodward (2001), but differ in parameter tuning and dust source representation (Woodward et al., 2022). Similarly, MIROC-ES2L is developed based on the MIROC model, both using the dust scheme from SPRINTARS (Spectral Radiation-Transport Model for Aerosol Species) (Takemura
et al., 2009).

In addition to the CMIP6 archive, we include three fully coupled simulations to assess the influence of dust emission parameterizations. These include a CESM simulation for 2004–2013 using the physically based dust scheme of Kok et al. (2014b) (hereafter the Kok scheme; CESM2-CAM-Kok) (Li et al., 2022a), and two simulations from the Energy Exascale Earth System Model (E3SM) for 1980–2014: one using the Zender scheme (E3SM2-Zender), and the other using the Kok scheme (E3SM3-
Kok) (Feng et al., 2022; Xie et al., 2025). The Zender and Kok schemes differ fundamentally in their representation of dust sources: the Zender scheme employs an empirical dust source function to shift emissions toward preferential regions, whereas the Kok scheme adopts physically based parameterizations of soil erodibility without the use of dust source functions. These paired CESM and E3SM simulations allow evaluations of the effect of dust emission schemes within a single model framework, and the performance of a common dust scheme between different models. It is important to note that other model-specific
factors may affect dust emission simulations in CESM and E3SM. For example, CESM2-CAM-Kok simulates dust as mineral components, while CESM2-CAM-Zender assumes spatially uniform dust properties without accounting for mineralogy (Li et al., 2024), potentially affecting aerosol optical properties and radiative feedback on meteorology. E3SM3 also includes various updates over E3SM2 that may alter wind fields and other meteorological variables relevant to dust emissions (Xie et al., 2025).

We also compare the ESMs with two aerosol reanalysis products: Modern-Era Retrospective Analysis for Research and Applications version 2 (MERRA2, 1980–2014) (Gelaro et al., 2017), and Japanese Reanalysis for Aerosol (JRAero, 2011–2017) (Yumimoto et al., 2017). Unlike the free-running, fully-coupled CMIP6 simulations driven by historical forcings, MERRA2 and JRAero compute dust emissions based on assimilated meteorological and land surface variables. Both reanalyses are generated using global aerosol transport models constrained by data assimilation of bias-corrected satellite aerosol observations.





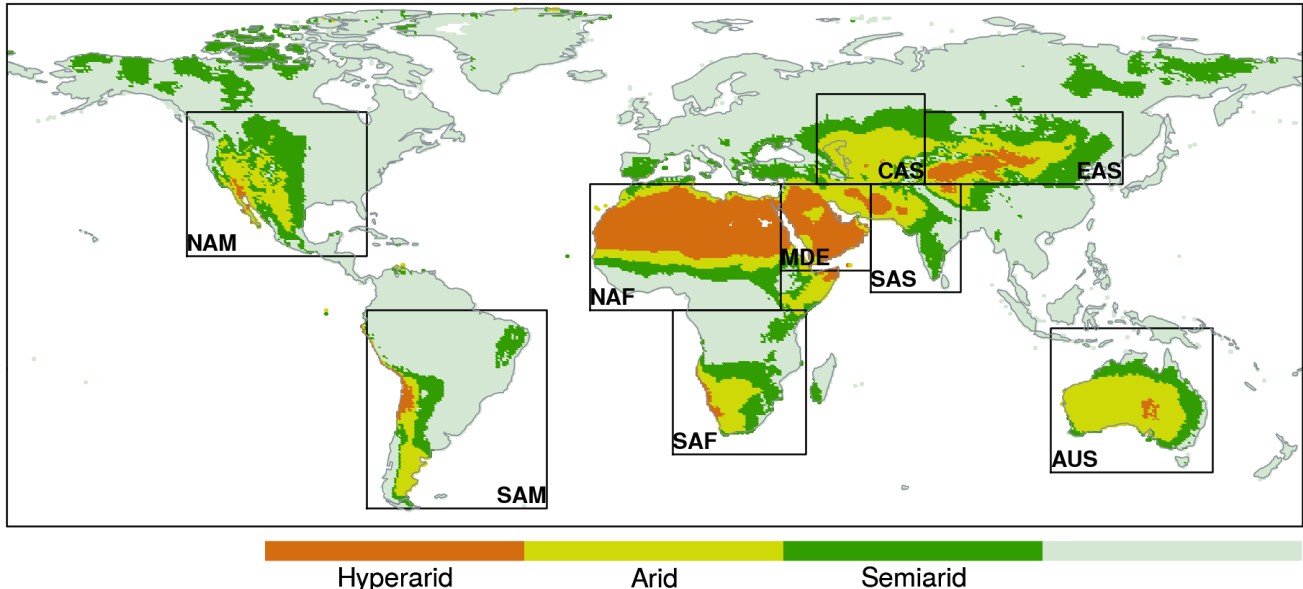

**Figure 1.** Definitions of nine geographic regions (boxes) and three climate zones (color shading) analyzed in this study. The nine regions are North Africa (NAF), Southern Africa (SAF), Middle East (MDE), Central Asia (CAS), East Asia (EAS), South Asia (SAS), Australia (AUS), North America (NAM), and South America (SAM).

MERRA2 utilizes the radiatively-coupled Goddard Chemistry, Aerosol, Radiation, and Transport (GOCART) module within GEOS-5 modeling system, in which dust emission is represented using the Ginoux et al. (2001) parameterization. JRAero, produced by the Japan Meteorological Agency, is based on the MASINGAR mk-2 global aerosol transport model and assimilates value-added MODIS AOD (Yumimoto et al., 2017). JRAero simulates dust emissions using the energy-based scheme of Shao et al. (1996), same as that used in the MRI-ESM2.0 aerosol model component (MASINGAR mk-2r4c) (Yukimoto et al.,
135    2019).

    We evaluate the climatological mean, spatial distribution, and interannual variability of dust emissions using monthly outputs from the ESMs and reanalysis products. We focus on nine geographic regions and three climate zones, as illustrated in Fig. 1. The geographic regions include North Africa (NAF), Southern Africa (SAF), Middle East (MDE), Central Asia (CAS), East Asia (EAS), South Asia (SAS), Australia (AUS), North America (NAM), and South America (SAM). The hyperarid, arid, and
semiarid climate zones are defined based on the aridity index (AI), calculated as the ratio of climatological precipitation to potential evapotranspiration over 1970–2000. Specifically, the hyperarid zone is defined by AI≤0.05, arid zone by 0.05<AI≤0.2, and semiarid zone by 0.2<AI≤0.5. As shown in Fig. 1, hyperarid regions cover most of North Africa, the Middle East, Iranian Plateau, and Tarim Basin. Arid and semiarid zones include other major dust sources, such as the Sahel (North Africa), Turan Depression (Central Asia), Gobi Desert (East Asia), Thar Desert (South Asia), Kalahari Desert (Southern Africa), Chihuahua
Desert (North America), Patagonia steppe (South America), and the Great Sandy and Simpson Deserts (Australia).





## 2.2 Dominance Analysis

Past studies have commonly used linear regression coefficients to quantify the influence of physical drivers on dust emission (e.g., Pu and Ginoux, 2016; Aryal and Evans, 2021; Zhao et al., 2022). In a multiple linear regression framework, the coefficient associated with a given predictor represents the mean change in the response variable per unit change in that predictor,

holding all other predictors constant. This interpretation, however, assumes mutual independence among predictors—an assumption that is often violated in dust studies, where dust emission drivers such as precipitation, soil moisture, and specific humidity are strongly correlated with each other. As a result, the regression coefficients may yield misleading inference of the predictor influence. Moreover, the regression coefficients, standardized or not, may not provide a consistent measure of predictor importance across models due to the varying dynamic ranges in different models.

In this study, we apply the dominance analysis technique to quantify the overall and relative influence of wind and hydroclimate drivers in modulating the dust variability within individual ESMs. The monthly dust flux is used as the target variable, while the following near-surface variables are used as predictors: 10-m wind speed, total precipitation (including both liquid and solid phases), water content in the uppermost soil layer (or soil moisture), 2-m specific humidity, 2-m air temperature, and leaf area index (LAI). Among the six predictors, wind speed represents the wind erosivity while the remaining variables

collectively represent the hydroclimate influence on the sediment availability. The selected hydroclimate variables are either directly used as input parameters in dust flux calculations or strongly correlated with the threshold wind velocity and dust emission intensity, as suggested in numerous studies (e.g., Engelstaedter et al., 2003; Ravi et al., 2004; Zou and Zhai, 2004; Lee and Sohn, 2009; Cowie et al., 2015; Xi and Sokolik, 2015a, b; Xi, 2023).

    Dominance analysis assesses the marginal contribution of each predictor to the total explained variance ($R^2$) by evaluating

all possible subset models within a multiple linear regression framework (Budescu, 1993; Azen and Budescu, 2003). With six predictors, this yields 63 possible combinations of predictors. For each predictor, the method calculates its average incremental contribution to the total $R^2$ across all subset models of the same size (i.e., models with the same number of predictors). These contributions are then averaged across all subset sizes to derive the predictor's unique contribution to the total $R^2$. A unique property of dominance analysis is that the sum of the individual predictor contributions equals the $R^2$ of the full model (i.e.,

with all predictors included), thus allowing for the partitioning of the total $R^2$ among a set of correlated predictors. The derived predictor $R^2$ can thus be interpreted as the portion of total variance in the dependent variable that is uniquely and jointly attributed to each predictor, accounting for the interactions and multicollinearity among the predictors.

    Dominance analysis is performed for all grid cells with nonzero dust emissions in ESMs and MERRA2 (JRAero is excluded due to missing predictors). The grid-level predictor $R^2$ values are then aggregated over predefined geographic regions and

climate zones (Fig. 1) to assess the model consistency in the collective and relative influence of the selected predictors. Prior to the analysis, annual cycles are removed from both dust emission fluxes and predictor variables. All variables are then normalized to a 0–1 range based on feature scaling to ensure equal weighting and comparability between the predictors. For ESMs that incorporate bare soil fraction as a scaling factor in dust flux calculations (e.g., INM-CM5.0, CNRM-ESM2.1, UKESM1.0), the dust fluxes are first normalized by the bare soil fraction to isolate the influence of the selected predictors.



**Figure 2.** Climatological mean dust emission fluxes from (a–u) individual ESMs, (v) model ensemble mean, (w) MERRA2 reanalysis, and (x) JRAero reanalysis. Global annual total dust emissions are displayed on each panel.



## 3 Results

### 3.1 Climatological mean

We begin by examining the climatological mean dust emission fluxes and comparing our results with previous assessments of AEROCOM and CMIP5 models. Figure 2 displays the mean dust fluxes from 21 ESMs, the model ensemble mean, and the MERRA2 and JRAero reanalysis, averaged over 2005–2014 for all models except CESM2-CAM-Kok (2004–2013) and JRAero (2011–2017). Most models capture the global dust belt stretching from West Africa to East Asia, along with the less intense sources in the Americas and Australia. E3SM3-Kok and HadGEM2-GC31 simulate the most spatially extensive dust-emitting areas, including high-latitude and subhumid regions. In contrast, CESM2-CAM-Zender, CESM2-WACCM-Zender, and NorESM2 simulate discrete and limited dust-emitting areas, due to the use of a truncated version of the Zender et al. (2003) dust source function, which excludes grid cells with values below 0.1. Unlike CESM2-CAM-Zender, E3SM2-Zender uses the original, untruncated dust source function and produces a more spatially continuous pattern (Fig. 2e).

Global annual total emissions vary greatly among the ESMs, ranging from 890 to 7727 Tg yr$^{-1}$, with nearly an order of magnitude difference (Fig. 2a–2u). The model ensemble mean estimate is 2786 Tg yr$^{-1}$ (Fig. 2v), with a standard deviation of 1821 Tg yr$^{-1}$, corresponding to a diversity of 65% (defined as the ratio of standard deviation to the model ensemble mean). Based on the 13 models which simulate particle diameters up to 20 $\mu$m, global annual emissions vary from 1062 to 6561 Tg yr$^{-1}$ with a mean of 3012 Tg yr$^{-1}$ and diversity of 51%. The uncertainty ranges are consistent with prior assessments. For example, Huneeus et al. (2011) compared 14 models from AeroCom Phase I and reported a global dust emission range of 500–4400 Tg yr$^{-1}$ with a diversity of 58%. Out of the 14 models, 7 models considered the diameter range of 0–20 $\mu$m and reported an emission flux of 980–4300 Tg yr$^{-1}$ with a diversity of 46%. Similarly, Gliß et al. (2021) compared 14 AeroCom Phase III models and found a range of 850–5650 Tg yr$^{-1}$ with a diversity of 64%. Wu et al. (2020) reported a range of 740–8200 Tg yr$^{-1}$ with a diversity of 66% across 15 CMIP5 models. Out of the 15 models, 7 models considering the diameter range of 0–20 $\mu$m yielded 740–3600 Tg yr$^{-1}$ with a diversity of 43%. More recently, Zhao et al. (2022) compared 15 models from the CMIP6 AMIP experiment and found a range of 1400–7600 Tg yr$^{-1}$ with a diversity of 61%. Collectively, these results along with our findings underscore persistent, substantial uncertainties in quantifying global dust emissions, despite improvements in model physics, parameterizations, and spatial resolutions over time.

The model ensemble mean global annual dust emission rate is significantly higher than that of MERRA2 (1605 Tg yr$^{-1}$, Fig. 2w), but closely aligns with JRAero (2780 Tg yr$^{-1}$, Fig. 2x). Overall, the model ensemble mean exhibits a more spatially continuous and homogeneous pattern than MERRA2 and JRAero. Particularly, over the Sahara Desert, the model ensemble mean simulates relatively evenly distributed emissions, whereas MERRA2 and JRAero display more localized and clustered patterns, possibly due to their topographic constraints on sediment erodibility. Compared to MERRA2 and JRAero, the model ensemble mean shows lower emissions over the western and central Sahara, but higher emissions over the Libyan Desert and Nile River basin.







**Figure 3.** Fractional contributions of dust emissions from different (a) geographic regions and (b) climate zones. Numbers indicate percentages above 5%.





## 3.2 Geographic distribution

Figure 3a presents the fractional contributions of nine geographic regions to global dust emissions. North Africa is identified across nearly all models as the largest source, contributing more than half of the global total. The model ensemble mean at-
tributes approximately 54% of emissions to North Africa, compared to 59% in MERRA2 and 41% in JRAero. Among the models, CanESM5.1 and INM-CM5.0 simulate relatively uniform emissions (Fig. 2i, 2q), with only one-third of emissions originating from North Africa, substantially lower than other models. These deviations likely reflect known deficiencies and errors in CanESM5.1 and INM-CM5.0. As noted in Sigmond et al. (2023), improper parameter tuning related to the hybridiza-tion of dust tracers induced spurious dust events in CanESM5.1, resulting in poor representations of dust distributions. In
addition, an interpolation error in the bare soil fraction distorted the model's dust source characterization, resulting in poor agreement with satellite observations (Sigmond et al., 2023) In INM-CM5.0, dust fluxes are calculated as a function of friction velocity only, without accounting for the influence of land surface conditions on the threshold wind velocity (Volodin and Kostrykin, 2016). While this simplified approach may be appropriate in persistently dry and barren regions, it may introduce substantial biases over regions where hydroclimate conditions strongly modulate dust emissions.

CESM2-CAM-Zender, CESM2-WACCM-Zender, and NorESM2 produce similar total emissions and regional fractions, suggesting that the choice of CAM vs. WACCM has minimal influence when the same dust scheme is used. The paired CESM and E3SM simulations with Zender and Kok schemes show large differences. Specifically, CESM2-CAM-Kok simulates 88% emissions from North Africa and the Middle East, significantly higher than CESM2-CAM-Zender (56%). Conversely, CESM2-CAM-Kok produces much less dust from East Asia (1% vs. 13% by CESM2-CAM-Zender), likely due to stronger soil moisture
suppression in the Kok scheme (Li et al., 2022b). Similarly, E3SM produces less emissions from Asia when using the Kok scheme.

GISS-E2 models exhibit minimal differences in the regional dust fractions between model versions (2.1 vs 2.2) and aerosol schemes (sectional OMA vs. modal MATRIX). However, total dust emissions are approximately 40% lower when using MA-TRIX compared to OMA, possibly due to differences in model tuning parameters. Also, as noted in Bauer et al. (2022), the
MATRIX modal size distribution underrepresent coarse dust particles (>5 $\mu$m diameter), which may contribute to lower dust emissions than OMA.

HadGEM3-GC3.1 and UKESM1.0 produce similar spatial distributions but differ in global totals by more than a factor of two. UKESM1.0, which is built on HadGEM3-GC3.1, uses the same dust scheme but includes several modifications that enhance the friction velocity and suppress the soil moisture, as described in Woodward et al. (2022). These model tunings
are expected to increase the surface gustiness and dryness in UKESM1.0, thereby enhancing its dust emission. Furthermore, UKESM1.0 excludes dust emission from seasonally vegetated areas, resulting in smaller dust-emitting areas (Fig. 2p) compared to HadGEM3-GC3.1 (Fig. 2o).

The three Japanese models (MRI-ESM2.0, MIROC-ES2L, and MIROC6) exhibit large differences in total emissions and, to a lesser extent, regional distributions. MRI-ESM2.0 produces nearly double the total emissions of JRAero, likely due to
differences between assimilated and free-running meteorological fields and/or in model tuning parameters. Despite using the





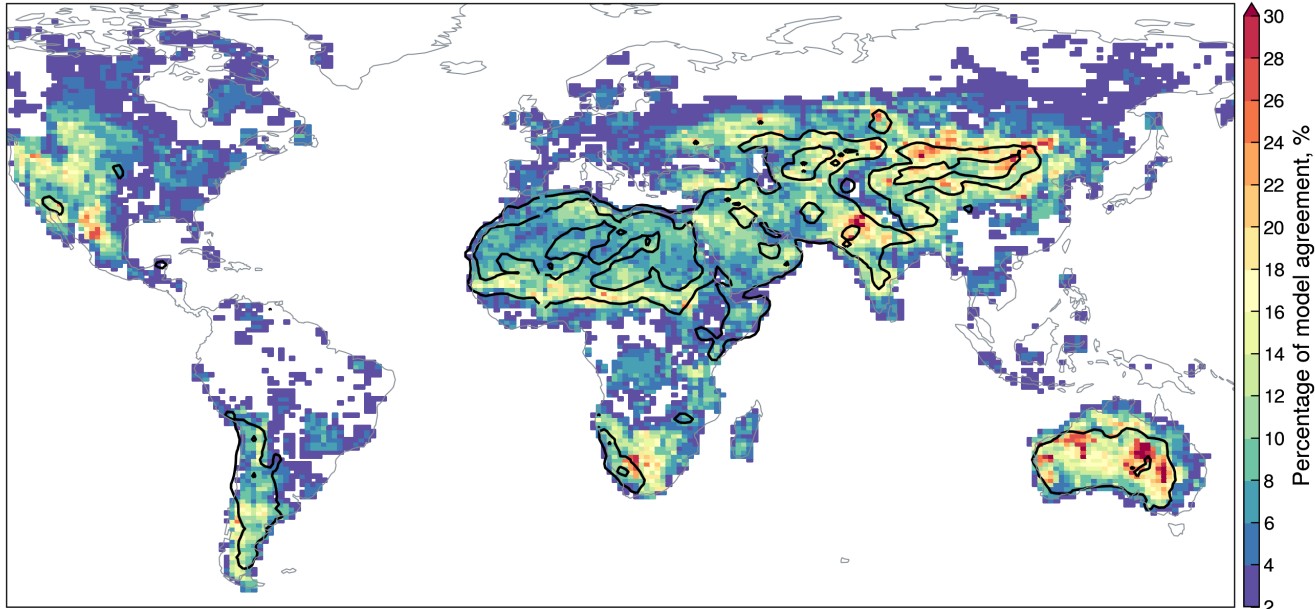

**Figure 4.** Percentage of statistically significant (p≤0.1) positive correlations out of every possible pairwise comparisons of monthly dust emission fluxes from 21 global models. Black contours represent the model ensemble mean annual dust flux of 10 and 100 Tg yr$^{-1}$.

same dust scheme, MIROC-ES2L produces five times more dust than MIROC6. This discrepancy is largely driven by stronger near-surface winds in MIROC-ES2L. We find that the global mean wind speed in MIROC-ES2L is about 50% higher than that in MIROC6. Furthermore, the prescribed LAI in MIROC6 is non-zero even in hyperarid regions, which likely contributes to lower emissions than MIROC-ES2L (Hiroaki Tatebe, personal communication).

Figure 3b shows the partitioning of global dust emissions among three climate zones, providing a first-order assessment of the dust sensitivity to hydroclimate conditions. Hyperarid regions, where dust emission is primarily controlled by wind speed, contribute between 41% (INM-CM5.0) and 88% (CESM2-CAM-Kok) of global emissions. Contributions from arid and semiarid regions show greater variability, in the range of 8–37% and 0–18%, respectively. Among the ESMs, CanESM5.1, INM-CM5.0, and UKESM1.0 exhibit the highest semiarid contributions at 18%, 15%, and 12%, respectively. Overall, the
model discrepancy increases with decreasing aridity, highlighting greater uncertainty over the transitional areas between dry and humid climates. This pattern mirrors the relatively lower model spread for North Africa and the Middle East, as shown in Fig. 3a, where the predominance of hyperarid regions leads to stronger model agreement in dust source attribution. Based on the model ensemble mean, global dust emissions are partitioned as 61% from hyperarid, 27% from arid, and 5% from semiarid zones. In comparison, MERRA2 (JRAero) produces 71% (65%), 26% (32%) and 3% (3%) of its global emissions
from hyperarid, arid, and semiarid regions, respectively.





## 3.3 Interannual variability

This section evaluates the degree of inter-model agreement in simulating the interannual variability of dust emissions. Monthly dust fluxes from the ESMs are first regridded to a common grid of 0.9°×1.25°, and deseasonalized by subtracting the month-wise climatological average at each grid cell. Spearman's rank correlation coefficients are then computed between the deseasonalized dust flux anomalies for all possible pairwise model comparisons. With 21 ESMs, this yields 210 unique pairwise comparisons. To quantify overall model agreement, we calculate the percentage of model pairs exhibiting statistically significant ($p \leq 0.1$) positive correlations. A higher percentage indicates stronger agreement among the models in simulating the year-to-year dust variability, whereas lower percentages reflect poor model agreement. The results are displayed in Fig. 4.

Despite being the largest dust sources, North Africa and the Middle East exhibit poor model agreement, with fewer than 12% of pairwise comparisons showing significant and positive correlations. This suggests that global models diverge substantially in their simulation of dust variability over these regions. Given that dust emissions from permanently dry, barren surfaces are predominantly controlled by wind speed, the poor agreement likely reflects inconsistent simulation of near-surface winds. Indeed, we find that only 10% of model pairs yield significant and positively correlated wind fields. Evan (2018) noted that dust-generating winds over the Sahara are mainly driven by large-scale meteorological processes, and that most CMIP5 models failed to capture the variability in surface winds. These findings highlight the critical role of low-level winds in improving dust model consistency and accuracy over hyperarid regions.

Interestingly, a slightly better inter-model agreement is observed over the Sahel, where dust emissions are more influenced by hydroclimate conditions. Most other arid/semiarid regions (particularly South Asia, East Asia and Australia) exhibit even better model consensus, indicating that a greater number of models simulate consistent dust variability in areas where emissions are strongly modulated by land cover and moisture availability. These results imply that the influence of hydroclimate conditions on dust emissions may be more consistently represented across models than wind extremes.

Figure 5 presents the correlation matrix between global and regional averaged dust flux anomalies from the ESMs and reanalysis products. The percentages of positive and negative correlations as well as their ratio are calculated to provide a measure of overall model agreement for each region. Globally, the model intercomparisons yield 56% positive correlations and 44% negative correlations (Fig. 5a). Based on a significance level of $p \leq 0.1$, however, only 14% comparisons are strongly positively correlated and 7% negatively correlated, corresponding to an agreement-to-disagreement ratio of 1.9. The highest correlation is found between MERRA2 and JRAero, suggesting that assimilated meteorological and land surface conditions exert a strong, shared influence on the dust variability in reanalysis products. The consistency between MERRA2 and JRAero is also observed for individual geographic regions (Fig. 5b–5i). Among the ESMs, only CNRM-ESM2.1, UKESM1.0, and GFDL-ESM4 show significant correlations with either MERRA2 or JRAero, while other models exhibit either negative or insignificant relationships with the reanalyses. Particularly, MRI-ESM2.0 shows no significant correlation with JRAero, despite using the same dust scheme. The CESM, E3SM, and GISS-E2 model families show generally poor internal agreement except between E3SM2-Zender and E3SM3-Kok. Despite the common model heritage, poor agreement is found between UKESM1.0 and HadGEM3-GC3.1, and between MIROC6 and MIROC-ES2L.



**Figure 5.** Spearman's rank correlation coefficients between global and regional monthly dust emission flux anomalies: (a) Global (GLB), (b) North Africa (NAF), (c) Southern Africa (SAF), (d) Middle East (MDE), (e) Central Asia (CAS), (f) South Asia (SAS), (g) East Asia (EAS), (h) North America (NAM), and (i) Australia (AUS). Dots indicate statistically significant correlations (p≤0.1). Summary tables are based on global models only (MERRA2 and JRAero not included).




The model intercomparisons for individual regions are summarized below, focusing on statistically significant relationships between global models and reanalysis products, and within the CESM, E3SM and GISS-E2 model families:

(1) North Africa shows poor model agreement, with only 10% of pairings positively correlated, below the global average. Only GFDL-ESM4 and CESM2-CAM-Zender correlate with the reanalyses. NorESM2 is even negatively correlated with MERRA2. CESM, E3SM and GISS-E2 models exhibit weak internal correlations.

(2) Southern Africa exhibits poor agreement, with only 8% positive correlations and an agreement-to-disagreement ratio of 1.1. CanESM5.1 is the only model positively correlated with MERRA2, while three models (MPI-ESM1.2, HadGEM3-GC3.1 and NorESM2) show negative correlations. The CESM, E3SM and GISS-E2 models exhibit poor internal agreement.

(3) Middle East reports 12% positive correlations and an agreement-to-disagreement ratio of 1.9. MPI-ESM1.2 and E3SM2-Zender correlate with the reanalyses. CESM, E3SM, and GISS-E2 model families show positive relationships between GISS-E2.1-MATRIX and GISS-E2.2-OMA as well as between CESM2-CAM-Zender and CESM2-CAM-Kok.

(4) Central Asia shows better model agreement with 13% positive correlations and an agreement-to-disagreement ratio of 3.1. E3SM3-Kok, GISS-E2.1-MATRIX and CanESM5.1 exhibit modest agreement with the reanalyses. E3SM2-Zender and E3SM3-Kok show consistent dust variability.

(5) South Asia exhibits the best model agreement, with 24% positively correlated pairings and an agreement-to-disagreement ratio of 6.2. 9 out of 21 models correlate with either MERRA2 or JRAero. GISS-E2 models show strong consensus. CESM2-CAM-Zender and CESM2-WACCM-Zender show moderate agreement.

(6) East Asia reports 10% positive correlations and the least (3%) negative correlations, with an agreement-to-disagreement ratio of 3.5. Six ESMs are positively correlated with MERRA2. CESM and GISS-E2 families show moderate internal agreement.

(7) North America yields 12% positive correlations and an agreement-to-disagreement ratio of 2.3. GISS-E2.1-MATRIX is the only model with significant correlation with the reanalyses. The CESM, E3SM, and GISS-E2 model families exhibit weak internal relationships.

(8) Australia reports 12% positive correlations and an agreement-to-disagreement ratio of 2.9. Five CMIP6 models correlate with either MERRA2 or JRAero. GISS-E2 models show mixed results, with both positive and negative correlations. CESM2-CAM-Zender and CESM2-WACCM-Zender exhibit a strong agreement with each other.

(9) South America exhibits poor model coherence with 10% positive correlations and an agreement-to-disagreement ratio of 1.5 (not shown). Only NorESM2 is strongly correlated with the reanalyses. CESM2-CAM-Zender shows modest agreement with CESM2-WACCM-Zender.

Based on the above analysis, hyperarid regions exhibit poor model agreement in the dust variability, whereas arid and semiarid regions show more coherent simulations. To further examine how model consistency varies with climate regime, Fig. 6 presents pairwise correlation matrices of deseasonalized dust fluxes for hyperarid, arid, and semiarid zones. The proportion of positively correlated models increases from 10% in hyperarid regions to 14% in arid regions and 17% in semiarid regions, indicating better model agreement over regions where dust emissions are sensitive to hydroclimate and land surface conditions. However, semiarid regions exhibit a greater number of negatively correlated model pairs (15%) than hyperarid (5%) and arid





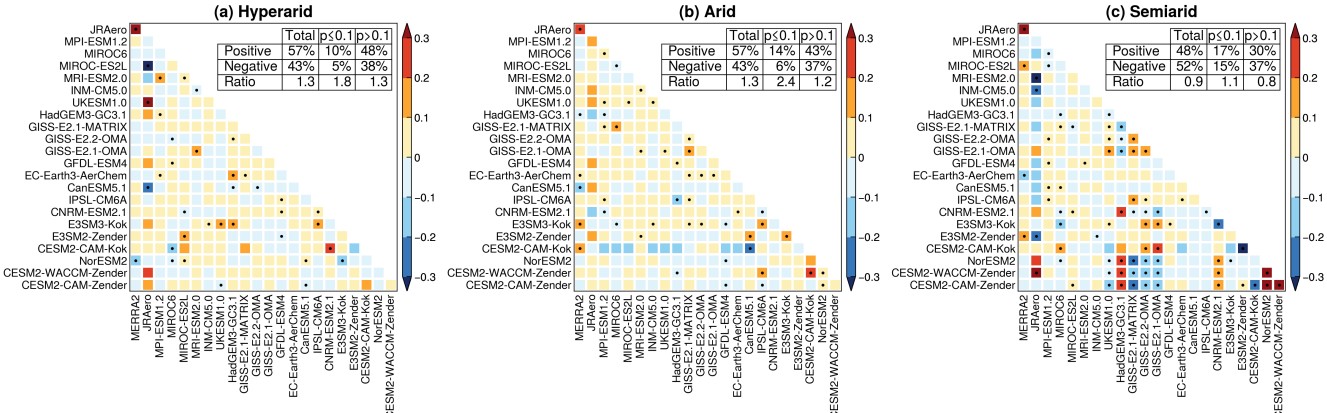

**Figure 6.** Same as Fig. 5 but for different climate zones.

(6%) regions. This dual pattern indicates that as the climate regime transitions from hyperarid to semiarid, a greater number of ESMs produce both consistent and divergent dust variability.

What causes such a complex behavior? In semiarid regions such as temperate grasslands and steppe ecosystems, dust emission is strongly modulated by antecedent land surface conditions, such as snow cover, soil moisture and vegetation growth-decay cycle, which exert strong lagged effects on dust activity in subsequent seasons (Shinoda et al., 2011; Nandintsetseg and Shinoda, 2015). Typically, dry anomalies during the prior wet season (e.g., early snow retreat or reduced rainfall) may delay vegetation onset or suppress vegetation growth, thereby extending the duration of bare soil exposure and enhancing the soil erodibility. This delayed dust response exemplifies the effect of land surface memory, where persistent land surface conditions influence subsequent climate processes. Therefore, we speculate that the simultaneous increase in model agreement and disagreement when transitioning from hyperarid to semiarid regions (Fig. 6c) reflects a "double-edged sword" effect of land surface memory: models with coherent representations of hydroclimate variability tend to converge in their dust emission simulations (i.e., more positive correlations), while those with divergent hydroclimate representations tend to diverge in dust responses (i.e., more negative correlations).

To verify the hypothesis, we examine the statistical association between inter-model correlations in dust emissions and those in hydroclimate conditions. Specifically, we perform principle component analysis (PCA) of the five hydroclimate variables (i.e., precipitation, soil moisture, specific humidity, temperature, LAI) for hyperarid, arid and semiarid zones. The first principle component (PC1), which accounts for at least 40% of total variances, is used to represent the dominant hydroclimate variability for each climate zone. Then, Spearman's rank correlation coefficients are computed for all pairwise comparisons of deseasonalized monthly PC1 values, similar to the dust emission flux comparisons in Fig. 6.

Figure 7 displays the correlation coefficients for model pairs with same-sign relationships (i.e., either both positive or both negative) in dust emissions and hydroclimate PC1. The regression slope and coefficient of determination ($r^2$) hence quantify the degree of statistical association between model consistencies in simulating the dust and hydroclimate variability. The





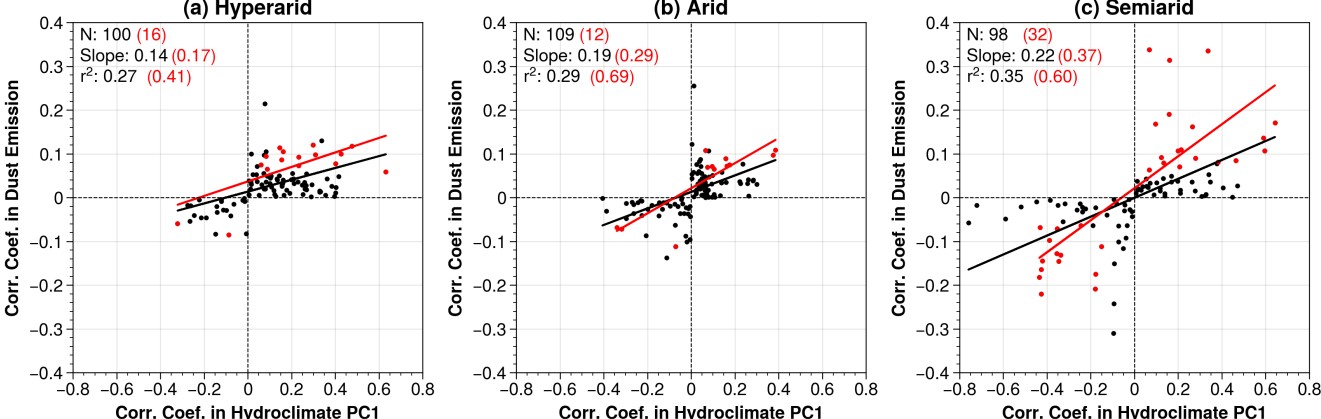

**Figure 7.** Statistical associations between pairwise correlation coefficients (p≤0.1 shown in red) in deseasonalized dust emission fluxes and hydroclimate variability averaged over (a) hyperarid, (b) arid, and (c) semiarid regions. Hydroclimate variability is represented by the first principle component (PC1) of five near-surface variables: precipitation, soil moisture, specific humidity, air temperature, and LAI.

positive association across all climate zones suggests that models with stronger agreement in hydroclimate conditions tend to produce more consistent dust variability, and vice versa. Furthermore, both the number of significantly correlated model pairs and the correlation strength (slope and $r^2$) increase progressively from hyperarid to semiarid zones, suggesting that 355 an increasing number of models simulate both coherent and incoherent dust and hydroclimate variability. This analysis hence provides support to our hypothesis regarding the dual role of land surface memory over semiarid regions: it enhances agreement among models with coherent hydroclimate representations, while simultaneously exacerbating disagreement among those with divergent hydroclimate variability.

### 3.4 Relative importance of wind and hydroclimate drivers

In this section, we present the dominance analysis of the collective and relative influence of wind and hydroclimate variables on the dust variability. Figure 8 presents the total $R^2$ for each ESM and MERRA2, representing the collective explanatory power of all six predictors. Results for CESM2-WACCM-Zender and NorESM2 are very similar to those of CESM2-CAM-Zender and thus not shown.

The models exhibit substantial differences in the total $R^2$, reflecting large variations of the internal model variability and 365 the coupling strength between dust emission and its physical drivers within individual ESMs. Among the models, CanESM5.1 yields the lowest $R^2$ globally, followed by MPI-ESM1.2, MIROC6, and EC-Earth3-AerChem, all of which show low $R^2$ values over extensive areas, suggesting that the selected predictors explain only a small fraction of the simulated dust variability in these models. The low explanatory power may be explained by several reasons. For example, known model errors and artifacts in CanESM5.1, as discussed in Section 3.1, may weaken or distort the relationships between dust emission and its physical 370 drivers. In models like INM-CM5.0, the use of over-simplified parameterizations or time-invariant input fields may weaken







**Figure 8.** The total explained variance ($R^2$) of monthly dust emission fluxes by six near-surface predictors in individual ESMs and MERRA2. Global mean $R^2$ values are shown on each panel.





**Figure 9.** The ratio of wind-associated $R^2$ to the combined $R^2$ of five hydroclimate variables (precipitation, soil moisture, specific humidity, air temperature, LAI) in individual ESMs and MERRA2.





the dust–predictor relationship. In addition, dust emission involves inherently nonlinear processes and its relationship with predictors may deviate from the linearity assumption in dominance analysis. As shown in Fig. 8, the total $R^2$ tends to be significantly lower in semiarid areas than hyperarid areas, likely due to the increased nonlinearity between dust emission and hydroclimate variables which diminishes their collective explanatory power. Finally, the use of monthly mean model output,
necessitated by data availability, may dampen the short-term variability and statistical association between the dust flux and predicting variables.

Despite the linearity assumption, most ESMs produce significant $R^2$ values over major dust-producing regions. This is particularly true for North Africa and the Middle East, where the total $R^2$ values exceed 0.6, indicating that the selected predictors capture a large portion of the simulated dust variability. Switching from the Zender to Kok scheme leads to generally
lower $R^2$ values in CESM globally, and in E3SM over most areas except Asia. GISS-E2 models show minor changes in the total $R^2$ when using the OMA or MATRIX microphysics scheme, and a modest increase between version 2.1 and 2.2. UKESM1.0 and HadGEM3-GC3.1 show minimal differences, both with high $R^2$ values globally. MIROC6 yields significantly lower $R^2$ than MIROC-ES2L, especially over hyperarid regions. MERRA2 yields significantly higher $R^2$ than most ESMs, especially over semiarid regions. Overall, the total $R^2$ patterns reflect large differences among ESMs and spatial inhomogeneity in the
collective contribution of the selected predictors to the dust variability.

To compare the relative importance of wind and hydroclimate drivers among the ESMs, Fig. 9 displays the ratio of the wind speed-associated $R^2$ to the combined $R^2$ of five hydroclimate variables (precipitation, soil moisture, specific humidity, air temperature, LAI). In all ESMs except GFDL-ESM4, the wind-to-hydroclimate $R^2$ ratio is well above 1 over hyperarid regions, consistent with the predominant control of wind speed on dust emissions from persistently dry, barren surfaces. In
contrast, arid and semiarid regions exhibit great variability, with the ratios above and below 1 depending on the model. This reflects increased model uncertainty or disagreement regarding the relative importance of wind and hydroclimate drivers in transitional regions where dust emission is more sensitive to land surface conditions.

Based on the wind-to-hydroclimate $R^2$ ratios, we classify global dust-emitting areas into three regimes: wind-dominated (ratio>1.2), hydroclimate-dominated (ratio<0.8), and equally-important (0.8–1.2). For each model, we calculate the fractions
of dust emissions originating from each regime. The results are displayed in Fig. 10. The ESMs show general consistency in the "equally-important" regime, with most ESMs simulating less than 10% of global emissions from regions where wind and hydroclimate drivers have nearly equal contributions. GFDL-ESM4 yields the highest contribution (12%) in this regime, while MERRA2 yields only 1%, lower than the ESMs.

Most ESMs and MERRA2 simulate more than 80% of dust emissions from the wind-dominated regime, consistent with
the dominant contribution of hyperarid areas to global dust production. However, three models deviate from this pattern. GFDL-ESM4 and INM-CM5.0 simulate less than 60% of emissions from wind-dominated regions, while CanESM5.1 yields a moderately lower fraction of 75%. Correspondingly, these models exhibit substantially higher contributions from hydroclimate-dominated regions than other models. The anomalous emission partitioning in these models may be explained by different reasons. In INM-CM5.0 and CanESM5.1, the more balanced partitioning may reflect their spatially homogeneous dust emission
patterns. In contrast, the behavior of GFDL-ESM4 is driven by anomalously weak wind and strong hydroclimate influence over





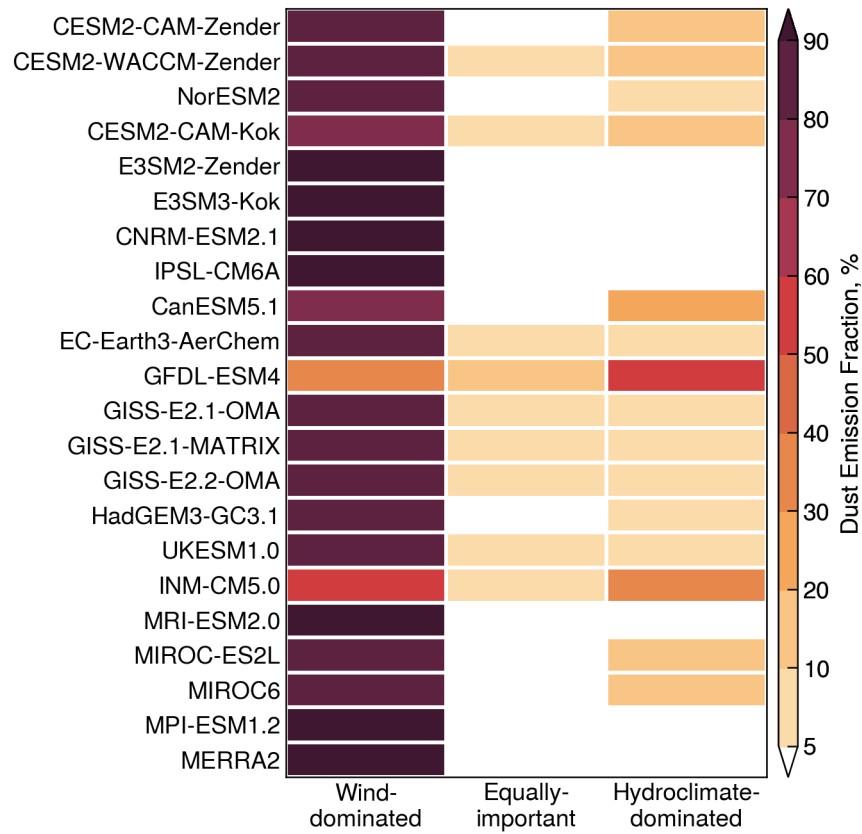

**Figure 10.** Fractional contributions of three regimes (wind-dominated, equally-important, and hydroclimate-dominated) to dust emissions in global models.

hyperarid regions. As shown in Fig. 9i, GFDL-ESM4 exhibits markedly lower wind-to-hydroclimate ratios over West Africa, the Middle East, and Iranian Plateau compared to other ESMs. Among the CESM and E3SM models, switching from the Zender to Kok dust scheme slightly reduces the wind-dominated emission fraction—from 85% to 80% in CESM, and from 99% to 96% in E3SM. GISS-E2 models show minimal sensitivity to model version or aerosol scheme, with emission fractions in the range of 82–85%. Similarly, UKESM1.0 and HadGEM3-GC3.1 yield identical estimates, with 90% of dust emitted from wind-dominated regions. MERRA2 simulates 98% emissions from wind-dominated regions, exceeding most ESMs.

Figure 10 confirms the anomalous dust distributions in CanESM5.1 and INM-CM5.0 as previously shown in Fig. 3, and further highlights GFDL-ESM4 as an outlier in representing the relative importance of wind and hydroclimate drivers. To further evaluate model consistency in predictor importance, we compute the fractional contribution of wind speed to the total $R^2$ across hyperarid, arid, and semiarid zones. The grid-level wind-associated $R^2$ fractions are illustrated in Fig. 11. A median fraction exceeding 50% indicates that wind speed accounts for the majority of explained dust variability at more than half of the grid cells.





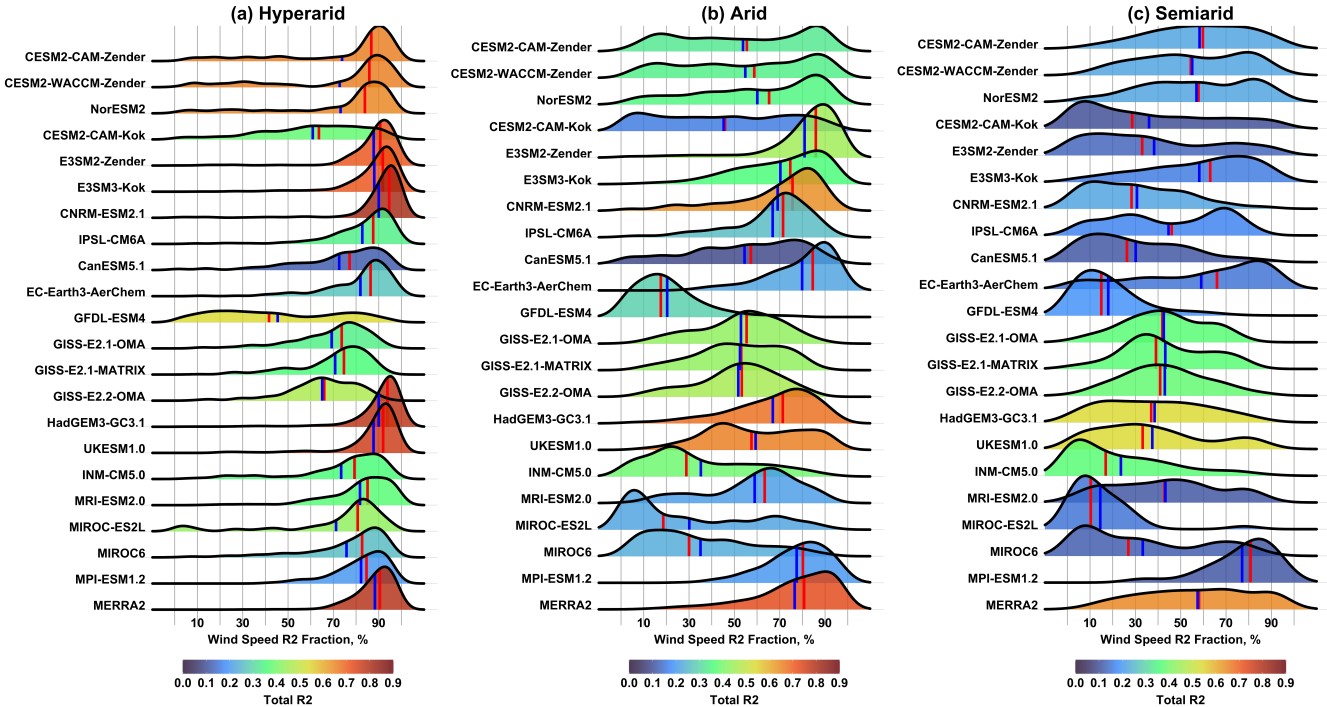

**Figure 11.** Ridgeline plots of grid-level fractional contributions of wind speed to the total $R^2$ over (a) hyperarid, (b) arid, and (c) semiarid regions. Red (blue) vertical lines indicate median (mean) values. Color shading represent the mean total $R^2$ of all predictors.

In hyperarid regions, most ESMs and MERRA2 capture the dominant wind control, with median wind $R^2$ fractions exceeding 80%. GISS-E2 models show slightly weaker wind influence (∼70%). In contrast, GFDL-ESM4 and CESM2-CAM-Kok exhibit substantially lower wind influence and greater spatial variability. In particular, GFDL-ESM4 yields a median wind $R^2$ fraction of 42%, suggesting that hydroclimate drivers dominate dust variability across most hyperarid grid cells. Within the CESM family, the choice of CAM vs. WACCM modules has a minor effect. However, replacing the Zender with Kok scheme led to a marked drop of total $R^2$, lower wind influence (64%), and increased spatial variability. These differences persist even when comparing common dust-emitting regions, suggesting that the differences arise primarily from the dust scheme. In contrast, switching from the Zender to Kok scheme in E3SM results in negligible differences. GISS-E2 models also show broad consistency between the OMA and MATRIX aerosol schemes, with a slight decline in wind-associated $R^2$ fraction from version 2.1 to 2.2.

In arid regions, the models show generally lower total $R^2$, indicating reduced explanatory power of the selected predictors. Wind speed remains the dominant driver in most ESMs and MERRA2, though its relative influence is diminished and more spatially heterogeneous. Four models—GFDL-ESM4, INM-CM5.0, MIROC-ES2L and MIROC6—yield median wind $R^2$ fractions well below 50%, signaling a transition from wind- to hydroclimate-dominated regimes. Despite large spatial variability, CESM2-CAM-Kok also reflects this transition, with a median wind $R^2$ fraction of 46%. In both CESM and E3SM,



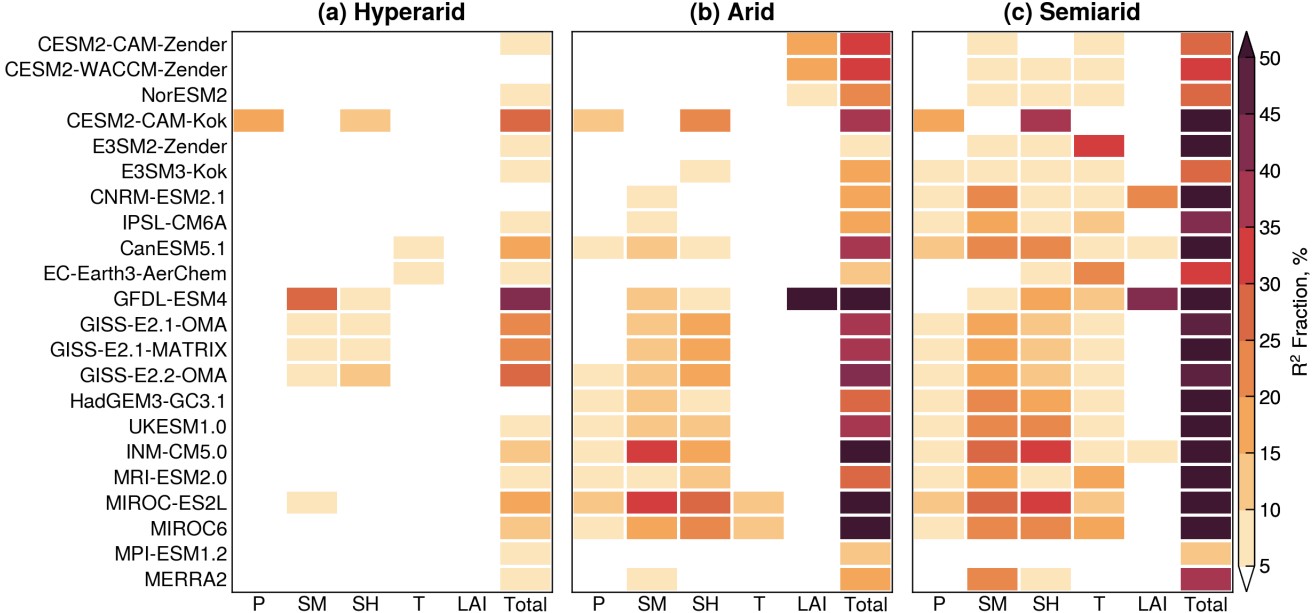

**Figure 12.** Median factional contributions of hydroclimate variables to the total $R^2$ in individual models over (a) hyperarid, (b) arid, and (c) semiarid regions. The hydroclimate variables are precipitation (P), soil moisture (SM), specific humidity (SH), air temperature (T), and leaf area index (LAI).

switching from the Zender to Kok dust scheme reduces the wind dominance. This reduction may be explained by the physically based soil erodibility treatment in the Kok scheme, which enhances dust sensitivity to hydroclimate conditions, as previously

435   suggested (Kok et al., 2014a).

In semiarid regions, the influence of wind speed further declines, while hydroclimate controls become increasingly important in all models. However, the extent and spatial pattern of this shift vary widely, leading to substantial model disagreement regarding the relative importance of wind and hydroclimate drivers. Specifically, hydroclimate dominate persists in CESM2-CAM-Kok, GFDL-ESM4, INM-CM5.0, MIROC-ES2L and MIROC6, while a transition from wind- to hydroclimate-dominated

440   regimes is observed in E3SM2-Zender, CNRM-ESM2.1, CanESM5.1, HadGEM3-GC3.1, and UKESM1.0. The remaining models (especially MPI-ESM1.2) and MERRA2 continue to show wind dominance, albeit with lower wind $R^2$ fractions and greater spatial variability. Interestingly, replacing the Zender with Kok dust scheme produces contrasting effects in CESM and E3SM, with reduced wind influence in CESM but enhanced wind influence in E3SM.

The above analysis identifies GFDL-ESM4 and CESM2-CAM-Kok as outliers in hyperarid regions due to their anomalously

445   strong hydroclimate influence. To investigate the source of this influence, Fig. 12 presents the median fractional contributions of individual hydroclimate variables to the total $R^2$ across different climate zones. Counterintuitively, CESM2-CAM-Kok exhibits elevated contributions from precipitation and specific humidity in hyperarid regions, where both variables are typically characterized by minimal variability. These influences further intensify over arid and semiarid regions. GFDL-ESM4 exhibits



strong sensitivity to soil moisture in hyperarid regions, and to LAI in arid and semiarid zones. All GISS-E2 models also display elevated sensitivity to soil moisture and specific humidity, which likely explains their moderate wind dominance in hyperarid regions (Fig. 11a). The anomalously strong hydroclimate influence in these models imply that they are likely to produce strong dust responses to hydroclimate variability or pattern changes.

## 4    Conclusions

This study evaluates the climatological distribution and interannual variability of historical dust emission fluxes simulated by 21 ESMs and two reanalysis products (MERRA2 and JRAero) across major geographic regions and climate zones (hyperarid, arid, and semiarid). The dominance analysis technique is applied to quantify the relative contributions of near-surface wind speed and five hydroclimate variables (including precipitation, soil moisture, specific humidity, air temperature, and LAI) to the simulated dust variability by each model. By treating dust emission flux as a model-specific quantity shaped by the internal variability and physical parameterizations within individual models, this study provides new insights into the consistencies, discrepancies, and biases in dust emission representations in ESMs.

Substantial model discrepancies are observed in both global total dust emissions and their regional distributions, consistent with findings from previous AEROCOM and CMIP5 model intercomparisons. While most ESMs identify North Africa and the Middle East as the dominant dust sources, CanESM5.1 and INM-CM5.0 simulate significantly lower contributions from these regions, possibly due to known model deficiencies and limitations. Furthermore, models exhibit poor agreement in simulating the interannual dust variability, especially over hyperarid regions where only 10% of the pairwise model comparisons yield statistically significant positive correlations. This reflects inconsistent representations of near-surface winds, the primary driver of dust emissions from permanently dry, barren surfaces. While most models capture the dominant control of wind speed in hyperarid regions, there are a few exceptions. Specifically, GISS-E2 models show slightly lower wind influence due to elevated sensitivity to soil moisture and specific humidity. GFDL-ESM4 and CESM2-CAM-Kok exhibit significantly lower wind influence and greater spatial variability, due to anomalous sensitivities to soil moisture in GFDL-ESM4 and to precipitation and specific humidity in CESM2-CAM-Kok.

In arid and semiarid regions where dust mobilization is increasingly influenced by vegetation cover and water availability, model behavior becomes more complex, exhibiting both enhanced consistency and divergence in the dust variability. Empirical analysis reveals that this behavior arises from a dual effect of land surface memory: models with coherent representations of hydroclimate variability tend to converge in their dust emission responses, while those with divergent hydroclimate representations diverge in dust variability. While all models capture the expected increase of hydroclimate influence with decreasing aridity, the extent and spatial pattern of this shift vary widely, leading to increased variability and discrepancy regarding the relative importance of wind and hydroclimate drivers. These findings have important implications for understanding and predicting dust responses to variations in atmospheric circulations, drought frequency, and land use practices, particularly over transitional regions where these factors strongly interact with each other in modulating the dust emission dynamics.



The effects of replacing the Zender et al. (2003) dust scheme with the new Kok et al. (2014b) scheme are also examined in CESM and E3SM. In CESM, the Kok scheme results in stronger hydroclimate influence and greater spatial variability across all climate zones. In E3SM, there are mixed responses in the predictor importance: negligible change over hyperarid regions, enhanced hydroclimate influence over arid regions, and increased wind influence over semiarid regions. While the exact causes of these changes are difficult to isolate due to concurrent model modifications (e.g., incorporation of dust mineralogy in CESM, model physics updates in E3SM3), the Kok scheme's physically based treatment of soil erodibility and removal of prescribed dust source functions likely contribute to enhanced hydroclimate influence and reduced wind influence on dust emissions, especially in arid and semiarid regions.

In summary, this study highlights substantial inconsistencies in how global models represent dust emission variability and its underlying physical drivers. While most models capture the dominant influence of near-surface winds in hyperarid regions, there is considerable divergence in arid and semiarid zones, where hydroclimate and land surface processes play an increasingly important role. Improving model representations of soil and vegetation dynamics and dust-climate interactions in these regions is thus essential for reducing uncertainties in predicting the dust response to climate variations and changes.

*Data availability.* Dust model comparison and dominance analysis results are available at https://doi.org/10.5281/zenodo.15741734.

*Author contributions.* XX designed this study with input from LL. XX and XL performed the analysis and wrote the initial manuscript. LL performed CESM2-CAM-Kok simulations. YF performed E3SM simulations. All authors edited the manuscript.

*Competing interests.* The authors declare no competing interests.

*Acknowledgements.* X.L. and X.X. are partially supported by the NASA Land-Cover and Land-Use Change Program. L.L. acknowledges support from the U.S. Department of Energy (DOE) under award DE-SC0021302, and from the Earth Surface Mineral Dust Source Investigation (EMIT), a National Aeronautics and Space Administration (NASA) Earth Ventures-Instrument (EVI-4) mission. He also acknowledges the high-performance computing resources provided by Derecho at the National Center for Atmospheric Research (NCAR), through NCAR's Computational and Information Systems Laboratory (CISL), which is sponsored by the National Science Foundation (NSF). The authors acknowledge the World Climate Research Programme for coordinating and promoting CMIP6, and thank the climate modeling groups for producing and making available their model output, the Earth System Grid Federation (ESGF) for archiving the data and providing access, and the multiple funding agencies who support CMIP6 and ESGF.





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
