# Peer review of "The relative importance of wind and hydroclimate drivers in modulating windblown dust emissions in Earth system models"

_EGUsphere, 2025_

## Referee Comment (RC2)

This article analyzed the inter-model discrepancies of the spatial distributions and interannual variability over dust source regions and climate zones, and quantified the total and relative explainability of wind speed and 5 hydroclimate parameters with the dominance analysis. The analyses of this study could help better understand the uncertainties of dust emissions among various Earth system models. The analyses and methods are reasonable. However, it would benefit from several improvements in terms of the size distribution and size cut of emitted dust, model resolution, model screening, and comparisons of dust burden and physical parameters against observations.

**General Comments**

1. As the simulated size distribution and size cut would largely affect the total dust emissions, it is recommended to better reconcile the total dust emissions for a defined size cut for a fair comparison and attributing the physical drivers. As shown in Table 1 in the study, there are large variations in the maximum diameter simulated in different Earth system models. In addition, different size distributions used would induce different total dust emissions even with the same model and dust emission scheme, but there is no information shown in this study about the size distribution of emitted dust.
2. As suggested in this study and various prior studies, different model resolutions could induce substantial and even orders of magnitude differences in total dust emissions. It is recommended to add the information of the model resolution used and potentially reconcile or better separate the differences induced by the model resolution.
3. It was suggested several times that some models may have intrinsic unreasonable representation in terms of dust emissions, for example in lines 218-223 and 368-370. Why are these models included in the inter-model comparisons for dust in this study? Would it make more sense to exclude them if they showed unreasonable representation of dust emissions?
4. What are the model performances in terms of total dust burden compared to ground-based and satellite-based observations? Why are the 5 hydroclimate parameters specifically chosen?
5. Out of curiosity, what are the annual trends of dust emissions and driving factors for different regions and climate zones? Are regions showing insignificant trend of total dust emissions the regions with larger interannual variability among models?

**Specific Comments**

1. For Table 1, what are the size distributions of emitted dust, and horizontal resolutions for different models?

2. Line 111-115: what the rationality of including these additional models or generally what is the criteria for selecting certain models for comparison?
3. Line 125-127: why are these two reanalysis products chosen? Are they treated as the reference? What are their model performance compared to observations?
4. Line 139-141: what is the data source of the aridity index? Are they climatological mean from a certain Earth system model? What are the annual trends of the aridity over these climate zones? Would semiarid zones shift to arid zones over the climatological range considered in this study?
5. Lin 173: are the target grid cells the same among different models? Do they vary as different models have different grid cells with nonzero dust emissions? If they vary, would the different number and location of target grid cells affect the intercomparisons?
6. Line187-190: why do these models use a truncated version of dust source function? How do they compare against observations for dust optical depth? If they show worse results, should they be excluded for the comparisons? As E3SM2-Zender uses the original dust source function and CESM2-Zender uses the truncated dust source function, this would be misleading for the conclusion about the effects of different dust emission schemes shown in line 481-487 as they are different Zender schemes.
7. Line 234-236: how about the size distribution in other models?
8. Line 247-249: how do the winds in MIROS-ES2L and MIROC6 compared to observations?
9. Line 263: what are their original resolutions?

---

## Author Comment (AC1)

We thank the reviewer for a speeding and thoughtful review. We appreciate the opportunity to clarify the novelty and distinct contributions of our study.

We agree with the reviewer that our study is a model inter-comparison. However, our approach differs substantially from previous studies. As the reviewer correctly pointed out, prior model intercomparison studies often rely on observational datasets (e.g., satellite-derived AOD) to evaluate or correlate with model simulations. While dust AOD (or other observations) offers some insight into the source strength, it primarily reflects the total atmospheric dust burden, which is influenced by transport and removal processes. Its limitations—such as retrieval challenges over bright desert surfaces, limited revisit frequency, and cloud interference—can further affect its ability to represent dust source activity.

In contrast, our study focuses on dust emissions only. As discussed in the Introduction, dust emission fluxes simulated by Earth system models are inherently a model-specific quantity, governed by internal processes such as dust emission parameterization, PBL, land surface and other coupled processes. Fig. 4 to Fig. 6 illustrate the inconsistent variability in dust emission simulations, highlighting a key motivation for our study. A key originality of our analysis lies in the examination of internal model drivers, rather than relying on external observational datasets. This approach enables us to probe the internal causal mechanisms within each model.

Moreover, our study introduces a novel spatial framework by partitioning the global domain into three distinct climate regimes. This approach reveals how model consistency varies systematically with climatic aridity. Particularly, the near-surface wind speed has been shown as the primary controlling factor over hyperarid regions, such as the Sahara. One might expect Earth system models to reproduce this feature reliably, however, our analysis reveals some models do not. Thus, despite being a model intercomparison, our analysis exposes fundamental biases in the dust emission representations in some models.

Another unique contribution of our study is the inclusion of a broader set of physical drivers than typically considered, combined with the use of dominance analysis to disentangle the individual contributions of correlated predictors. This method accounts for multicollinearity among drivers and quantitatively attributes their relative importance in explaining dust variability.

Finally, our study incorporates a wider range of Earth system models than previous studies. In particular, we include E3SMv2 and E3SMv3, which have not been compared within the CMIP6 framework, and an updated version of CESM2 with revised dust emission scheme and mineralogy. The paired model experiments with Zender and Kok schemes in E3SM and CESM allow us to examine how the same dust emission scheme performs in different ESM, and how different dust emission schemes perform in the same ESM.

We hope this response clarifies the key distinctions and contributions of our study. Below, we provide specific replies to the reviewer's additional comments.

This study investigates the dominant factors contributing to the spatial (regional and aridity-level dependent) and temporal (interannual) variability of dust emission, drawing upon Earth System

Model (ESM) results for the present day provided to CMIP6. Similar prior works (not cited in the manuscript) have previously reached comparable conclusions. So the work lacks originality. A key distinction is that previous studies relied on observations, allowing for absolute comparisons. This study, however, does not utilize global long-term observations of dust properties, such as satellite-based Dust Optical Depth (DOD). Consequently, its findings are limited to a model inter-comparison, without the ability to assess potential strengths or weaknesses.

I would suggest to the authors to build upon the work of Zender and Kwon (2005) and Kim et al. (2017), perhaps focusing on long-term variability, given that MODIS offers over 20 years of daily global DOD data. Surface concentration at Barbados has been observed daily since 1965, providing 60 years of data—ample for studying inter-annual variability. Prospero and Lamb (2003) demonstrated that hydroclimate factors control dust long-term variability. By examining the ESM results that best reproduce the interannual variability observed at Barbados and/or in satellite data, we could discern the strengths and weaknesses of their controlling factor(s).

However, when analyzing interannual variations, certain factors influencing these variations should also be considered. These include land-use, which significantly contributes to dust emission (Ginoux et al., 2012; Stanelle et al., 2014), and fires (Yu and Ginoux, 2022; Wagner and Schepanski, 2025).

Some fundamental information regarding the ESMs, crucial for understanding their differences in dust emission, is either incorrect or inadequately explained. Furthermore, the analysis exhibits a bias towards CESM, with minimal or no discussion of other models.

Response: The Introduction section aims to give a broad overview of inconsistent dust emission representations in ESMs, as a main motivation for our inter-comparison approach. We may have misinterpretations or miss important details .Can you please clarify what we may have missed?

The citations for MPI-ESM-1.2 and GFDL-ESM4 are erroneous. Tegen et al. (2019) described ECHAM6.3-HAM2.3 with constant roughness and vegetation mask. Is Mauritsen et al. (2019) not the correct reference for MPI-ESM-1.2? MPI-ESM-1.2 utilizes MAC-v1 prescribed aerosol distribution (Kinne et al., 2013). For GFDL-ESM4, it would be appropriate to refer to Shevliakova et al. (2024) instead of Evans et al. (2016). The authors are advised to consult these references or contact the lead authors to ensure an accurate description of their models.

Response: Thanks for pointing out the correct references. When choosing the references, we try our best to use the paper with most details on dust emission representations. We will update the references based on your recommendations.

Table 1 lacks critical information, such as LAI. Is it calculated online? Is it static or dynamic? Does it incorporate land-use? Is brown vegetation included? For 10-meter wind-speed derived from the first model level, the robustness of the derivation diminishes with increasing altitude of

this level. Horizontal resolution is paramount for all fields. How can models be compared without knowledge of their spatial resolution?

Response: Thanks for pointing out the additional model differences. We will add more details in the Table to highlight the dust emission differences. As the reviewer pointed out, the models are different in many aspects. This is exactly the reason why we approach the inter-comparison by treating dust emission as a model-specific quantity, and focusing on the internal variability and predictors, instead of using external observations.

Given these significant issues, I cannot recommend the publication of the present manuscript.

---

## Author Comment (AC2)

We thank the reviewers for their thoughtful and constructive comments. We made substantial changes to improve the overall structure of the paper and clarify the study's objective and novelty compared with previous studies. Detailed responses to the reviewers' comments are provided below.

**Reviewer #1**

This study investigates the dominant factors contributing to the spatial (regional and aridity-level dependent) and temporal (interannual) variability of dust emission, drawing upon Earth System Model (ESM) results for the present day provided to CMIP6. Similar prior works (not cited in the manuscript) have previously reached comparable conclusions. So the work lacks originality. A key distinction is that previous studies relied on observations, allowing for absolute comparisons. This study, however, does not utilize global long-term observations of dust properties, such as satellite-based Dust Optical Depth (DOD). Consequently, its findings are limited to a model inter-comparison, without the ability to assess potential strengths or weaknesses.

Response: Unlike previous studies, our work provides a more focused and in-depth analysis on dust emission. We evaluate how the simulated interannual variability of dust emission differs across CMIP models, which has not yet been addressed in the literature. The results of this work thus provide critical insights into how dust emission projections would differ across these models under future climate change. A novel aspect of this study is that dust emission is viewed as an unobservable, model-specific quantity, instead of a true, observable geophysical quantity, reflecting the lack of direct observational constraints on the dust emission magnitude. Multiple linear regression, which was used in past studies, cannot disentangle the influence of various dust emission drivers due to multicollinearity. Here we used the dominance analysis technique to quantify the joint and relative importance of correlated drivers. Finally, we include CESM and E3SM models with newly updated dust schemes, which remove the use of dust source functions and incorporate more physically based parameterizations, compared to models participating in the CMIP experiments. By comparing results from these new models, we demonstrate, for the first time, how the updated emission scheme simulates dust variability and sensitivities to the physical drivers. All these represent the originality of this work compared to past studies.

In response to reviewer's suggestions, we have attempted to use MODIS dust AOD (DAOD) for model evaluation. Unfortunately, the results are not encouraging, presumably because MODIS aerosol observations are not a good proxy for dust emission fluxes. See details below.

I would suggest to the authors to build upon the work of Zender and Kwon (2005) and Kim et al. (2017), perhaps focusing on long-term variability, given that MODIS offers over 20 years of daily global DOD data. Surface concentration at Barbados has been observed daily since 1965, providing 60 years of data—ample for studying inter-annual variability. Prospero and Lamb (2003) demonstrated that hydroclimate factors control dust long-term variability. By examining the ESM results that best reproduce the interannual variability observed at Barbados and/or in satellite data, we could discern the strengths and weaknesses of their controlling factor(s).

Response: This study's objective to evaluate the representation of interannual variability in dust emission within CMIP models. The surface concentration at Barbados is influenced by numerous factors beyond dust emission, such as those affecting deposition rates and transport pathways. Consequently, this variable cannot yield a definitive conclusion regarding dust emission itself. Nevertheless, we performed a new analysis by using MODIS deep blue aerosol products to evaluate model representations of the relative influence of dust emission drivers. The analysis is conducted in a consistent manner as for the models. Specifically, we performed dominance analysis using MODIS monthly DAOD (2001-2024) as target variable, and MERRA2 wind speed, precipitation, soil moisture, specific humidity, air temperature, and LAI as predictors. The MODIS DAOD is derived as follows: we first derive daily DAOD using the approach of Li and Ginoux (2025) for both Terra and Aqua MODIS level-2 products. The Terra and Aqua DAODs are averaged for the same day and regrided to 0.65°x0.5° resolution (same as MERRA2) and monthly levels. To isolate dust source regions, we focus on grid cells with at least 14 days of DAOD larger than 0.2, based on the frequency-of-occurrence method in Ginoux et al. (2012).

In Fig. (a) below, the total R² or variance explained by the six predictors is very low (<0.3) globally, substantially lower than R² values (>0.8) in MERR2 dust emissions as shown in Fig. 7t from the paper. This indicates weak coupling between DAOD and the reanalyzed wind and hydroclimate variables, suggesting that MODIS DAOD is not a good proxy for dust emission fluxes. Furthermore, Fig. (b) below shows the ratio of wind speed-associated R² to hydroclimate-associated R², calculated the same way as in Fig. 8 from the paper. We find that North Africa and Arabian Peninsula exhibit extensive areas of R² Ratio

However, when analyzing interannual variations, certain factors influencing these variations should also be considered. These include land-use, which significantly contributes to dust emission (Ginoux et al., 2012; Stanelle et al., 2014), and fires (Yu and Ginoux, 2022; Wagner and Schepanski, 2025).

Response: We included the leaf area index (LAI), an indicator of green vegetation structure and density (which indirectly reflects the land use state), as one of the predictors to account for the effect of vegetation and land-use. We did not aim to evaluate the model performance in capturing the LAI effect, but whether the models have consistent representations of the LAI effect.

Some fundamental information regarding the ESMs, crucial for understanding their differences in dust emission, is either incorrect or inadequately explained. Furthermore, the analysis exhibits a bias towards CESM, with minimal or no discussion of other models.

The citations for MPI-ESM-1.2 and GFDL-ESM4 are erroneous. Tegen et al. (2019) described ECHAM6.3-HAM2.3 with constant roughness and vegetation mask. Is Mauritsen et al. (2019) not the correct reference for MPI-ESM-1.2? MPI-ESM-1.2 utilizes MAC-v1 prescribed aerosol distribution (Kinne et al., 2013). For GFDL-ESM4, it would be appropriate to refer to Shevliakova et al. (2024) instead of Evans et al. (2016). The authors are advised to consult these references or contact the lead authors to ensure an accurate description of their models.

Response: We have updated the descriptions of ESMs and the references for dust schemes and models in Table 1 and the body text.

Table 1 lacks critical information, such as LAI. Is it calculated online? Is it static or dynamic? Does it incorporate land-use? Is brown vegetation included? For 10-meter wind-speed derived from the first model level, the robustness of the derivation diminishes with increasing altitude of this level. Horizontal resolution is paramount for all fields. How can models be compared without knowledge of their spatial resolution?

Response: We added a new column for horizontal resolutions. To not make Table 1 too big, we removed the details of soil moisture and vegetation treatments, but added new columns for LAI treatment and dust scheme references. Some models (e.g., NorESM2, CanESM5.1) use dynamic vegetation simulations, while others (e.g., GISS-E2.1, HadGEM3-GC31-MM) use prescribed LAI. The different treatments in vegetation, land-use, winds, and other factors are the primary reasons why dust emission flux should be considered as a model-specific quantity.

Given these significant issues, I cannot recommend the publication of the present manuscript.

**Reviewer #2**

This article analyzed the inter-model discrepancies of the spatial distributions and interannual variability over dust source regions and climate zones, and quantified the total and relative explainability of wind speed and 5 hydroclimate parameters with the dominance analysis. The analyses of this study could help better understand the uncertainties of dust emissions among various Earth system models. The analyses and methods are reasonable. However, it would benefit from several improvements in terms of the size distribution and size cut of emitted dust, model resolution, model screening, and comparisons of dust burden and physical parameters against observations.

Response: We would like to emphasize that, unlike previous multi-model studies, our analysis provides a more focused and in-depth assessment of dust emissions rather than the entire dust life cycle. Specifically, we quantify how the simulated interannual variability of dust emission differs across CMIP models - an aspect that, to our knowledge, has not been systematically

examined before. This focus enables us to identify the physical drivers of dust emission variability and provide new insights into how dust emission projections may diverge among ESMs in response to future climate changes.

Because dust emission is an unobservable quantity, we explicitly treat it as such rather than as an observable geophysical field, a distinction that is essential given the absence of direct, large-scale observational constraints. We acknowledge that there are multiple high-quality dust observations (e.g., AOD, surface concentration) other than dust emission fluxes. However, we cannot draw solid, clear conclusions on dust emission itself, particularly its temporal variability, considering that these observations are influenced by numerous factors beyond emissions, such as those affecting dust deposition rates, transport pathways, and atmospheric lifetime.

Please see below our detailed response to other comments.

**General Comments**

1. As the simulated size distribution and size cut would largely affect the total dust emissions, it is recommended to better reconcile the total dust emissions for a defined size cut for a fair comparison and attributing the physical drivers. As shown in Table 1 in the study, there are large variations in the maximum diameter simulated in different Earth system models. In addition, different size distributions used would induce different total dust emissions even with the same model and dust emission scheme,

but there is no information shown in this study about the size distribution of emitted dust.

Response: The models typically calculate a total emission flux (mass per area per time) using a vertical dust flux equation as a function of wind speed, soil moisture, and other soil erodibility factors. The total emission flux is then partitioned into discrete size bins based on assumed emission size distributions. That is, the emitted dust size distribution is not dynamically predicted in the models. The assumed size distributions and upper cutoffs can lead to large differences in the total mass flux, but have limited effects on the interannual variability of the simulated dust emission fluxes, especially how they respond to different drivers represented in the CMIP models (the focus of this study).

We have added below to the revised manuscript (L185).

We use the monthly total dust emission flux as target variable and consider six predictors, including 10-m wind speed, total precipitation (including liquid and solid phases), water content in the uppermost soil layer (hereafter as soil moisture), 2-m specific humidity, 2-m air temperature, and leaf area index (LAI). Here the total dust emission flux is a bulk quantity that represents the source strength. Although ESMs differ in the representation of emitted dust size distribution-a key factor influencing the transport and atmospheric lifetime of dust particles-we expect the particle size partitioning to have limited impact on diagnosing the emission process, particularly its dependence on wind and land surface conditions. The primary drivers of emission variability operate upstream of the size partitioning of the mobilized soil particles.

2. As suggested in this study and various prior studies, different model resolutions could induce substantial and even orders of magnitude differences in total dust emissions. It is recommended to add the information of the model resolution used and potentially reconcile or better separate the differences induced by the model resolution.

Response: We have added model resolutions in Table 1. The models differ in many aspects, such as model physics, dust schemes, parameter uncertainties, and configurations (i.e., horizontal resolution, vertical levels). Together, these differences contribute to inconsistent simulations of dust interannual variability and predictor influence. Separating the effect of model resolution would require experiments using the same model but with different resolutions, which is beyond the scope of this study.

3. It was suggested several times that some models may have intrinsic unreasonable representation in terms of dust emissions, for example in lines 218-223 and 368-370. Why are these models included in the inter-model comparisons for dust in this study? Would it make more sense to exclude them if they showed unreasonable representation of dust emissions?

Response: We aim to include as many models as possible in order to provide a comprehensive assessment of state-of-the-art Earth system models. Because there is no 'ground truth' or validation data for dust emission fluxes, it's almost impossible to determine which model performs the best. Through a multi-model comparison, however, we may be able to identify so-called outliers, for example, models which overestimate hydroclimate influences in hyperarid areas. The intrinsic unreasonable representations mentioned by the reviewer was already known in the community, including the model developers. Here, we just kept it as a sidenote.

4. What are the model performances in terms of total dust burden compared to ground-based and satellite-based observations? Why are the 5 hydroclimate parameters specifically chosen?

Response: We only compared total dust emission fluxes in this study. Evaluating model performance against dust concentration or AOD observations would require separate efforts and are not the focus in this work (please refer to our response above the general comments). We choose the five hydroclimate parameters, because they are either directly used in dust flux calculations or strongly correlated with dust emissions. The choice of predictors is described at L190 of the revised paper.

In response to Reviewer #1, we made an attempt to use MODIS deep blue aerosol products to evaluate model representations of dust emissions. However, the results are not encouraging and we decide to not include them in the revision.

We performed dominance analysis using MODIS monthly DAOD (2001-2024) as target variable, and MERRA2 wind speed, precipitation, soil moisture, specific humidity, air temperature, and LAI as predictors. The MODIS DAOD is derived as follows: we first derive daily DAOD using the approach of Li and Ginoux (2025) for both Terra and Aqua MODIS level-2 products. The Terra and Aqua DAODs are averaged for the same day and regrided to 0.65°x0.5° resolution (same as MERRA2) and monthly levels. To isolate the dust sources, we focus on grid cells with at least

14 days of DAOD larger than 0.2, based on the frequency-of-occurrence method in Ginoux et al. (2012).

In Fig. (a) below, the total R² or variance explained by the six predictors is very low (<0.3) globally, substantially lower than the R² values (>0.8) in MERR2 dust emissions as shown in Fig. 7t from the paper. This indicates weak coupling between DAOD and the reanalyzed wind and hydroclimate variables, and suggests that MODIS DAOD is not a good proxy for dust emission. Furthermore, Fig. (b) below shows the ratio of wind speed-associated R² to hydroclimate-associated R², calculated the same way as in Fig. 8 from the paper. We find that North Africa and Arabian Peninsula exhibit extensive areas of R² Ratio

5. Out of curiosity, what are the annual trends of dust emissions and driving factors for different regions and climate zones? Are regions showing insignificant trend of total dust emissions the regions with larger interannual variability among models?

Response: We did not look at dust trends but focused on dust emission's relationships with physical drivers. The recent study by Kok et al. (2023) found that CMIP6 models produced no significant trends in global dust burden from 1850 to 2000, and suggested that the models failed to reproduce the increased dust burden since preindustrial times, likely because of the model failure to capture historical climate and land-use drivers and/or the dust sensitivity to the drivers. This study addresses the latter issue, that is, the sensitivity of dust emissions to the driving factors.

1. For Table 1, what are the size distributions of emitted dust, and horizontal resolutions for different models?

Response: We have added model resolutions in Table 1. Because our analysis uses total dust emission fluxes, we only show the size upper cutoff in the table.

We have added below to the revised manuscript (L185).

"the total dust emission flux is a bulk quantity that represents the source strength. Although ESMs differ in the representation of emitted dust size distribution---a key factor influencing the transport and atmospheric lifetime of dust particles---we expect the particle size partitioning to have limited impact on diagnosing the emission process, particularly its dependence on wind and land surface conditions. The primary drivers of emission variability operate upstream of the size partitioning of mobilized soil particles."

2. Line 111-115: what the rationality of including these additional models or generally what is the criteria for selecting certain models for comparison?

Response: We did not have specific selection criteria, but tried to include as many models as possible in the inter-comparison to provide a comprehensive assessment of state-of-the-art Earth system models. This would help identify outliers or models with different behaviors from the majority. We also incorporated the newly updated CESM and E3SM models because they have implemented revised dust emission schemes that eliminate prescribed source functions and represent emission processes more physically. By comparing paired versions of these models, we wanted to demonstrate, for the first time, how the updated schemes modify dust emission responses to wind speed and hydroclimate drivers.

3. Line 125-127: why are these two reanalysis products chosen? Are they treated as the reference? What are their model performance compared to observations?

Response: JRAero and MERRA2 are both aerosol reanalysis products. We choose them because they provide data on monthly total dust emission fluxes, same as CMIP6 models. The reanalyzed dust emission fluxes are not treated as 'truth' or reference, because they are produced by global aerosol models, similar to CMIP6 models. The key difference is that the meteorological, land surface processes, and total aerosol information (e.g., aerosol optical depth) in the reanalysis are constrained by observational data assimilations. But still, dust emissions are simulated with prescribed dust sources and parameterizations. JRAero and MERRA2 also assimilate satellite-derived AOD, which constrains the dust column burden but has limited influences on dust emissions, which has no direct observational constraints.

4. Line 139-141: what is the data source of the aridity index? Are they climatological mean from a certain Earth system model? What are the annual trends of the aridity over these climate zones? Would semiarid zones shift to arid zones over the climatological range considered in this study?

Response: The aridity index is calculated from the 1970–2000 climatological mean precipitation and potential evapotranspiration provided by Zomer et al., (2022). We choose this approach to ensure that climate zones are consistent for all models, allowing a direct comparison of the same dust-emitting regions. We expect the climate aridity to have slight changes from year to year due to precipitation variations. But the aridity index is normally defined using a multi-decadal climatology. We use the climatological aridity index to keep the geographic coverage unchanged while comparing dust emissions from each climate zone.

5. Lin 173: are the target grid cells the same among different models? Do they vary as different models have different grid cells with nonzero dust emissions? If they vary, would the different number and location of target grid cells affect the intercomparisons?

Response: Yes, the number of target grid cells differs among the models, due to different resolutions and dust source area coverage (as shown in Fig. 2). In the inter-comparison, we used the median value of grid cells from a climate zone (Fig. 9) or the statistical distributions of the grid cells (Fig. 10).

6. Line187-190: why do these models use a truncated version of dust source function? How do they compare against observations for dust optical depth? If they show worse results, should they be excluded for the comparisons? As E3SM2-Zender uses the original dust source function and CESM2-Zender uses the truncated dust source function, this would be misleading for the conclusion about the effects of different dust emission schemes shown in line 481-487 as they are different Zender schemes.

Response: Dust models use different dust source functions to reflect the geomorphic, topographic or hydrologic constraints on the spatial pattern of dust emissions. The original Zender scheme uses a continuous dust source function (Zender et al., 2003). When the scheme was implemented in CESM2, the dust source function was truncated for values below 0.1, resulting in discontinuous emission patterns. In contrast, E3SM2-Zender uses the original, continuous source function. While comparing the models, e.g., between CESM2-CAM-Zender and CESM2-CAM-Kok, we used two strategies: all grid cells from each model, and using only common dust-emitting grid cells. We found that the results are very similar.

7. Line 234-236: how about the size distribution in other models?

Response: We have added below to the revised manuscript (L185).

"the total dust emission flux is a bulk quantity that represents the source strength. Although ESMs differ in the representation of emitted dust size distribution---a key factor influencing the transport and atmospheric lifetime of dust particles---we expect the particle size partitioning to have limited impact on diagnosing the emission process, particularly its dependence on wind and land surface conditions. The primary drivers of emission variability operate upstream of the size partitioning of mobilized soil particles."

8. Line 247-249: how do the winds in MIROS-ES2L and MIROC6 compared to observations?

Response: We did not compare the modeled winds with observations. We compared the MIROC-ES2L and MIROC6 winds in order to understand the causes of their dust emission differences.

9. Line 263: what are their original resolutions?

Response: We have added model resolutions to Table 1.

**References**

Ginoux, P., Prospero, J. M., Gill, T. E., Hsu, N. C., & Zhao, M. (2012). Global-scale attribution of anthropogenic and natural dust sources and their emission rates based on MODIS Deep Blue aerosol products. *Reviews of Geophysics*, *50*(3).

Kok, J. F., Storelvmo, T., Karydis, V. A., Adebiyi, A. A., Mahowald, N. M., Evan, A. T., ... & Leung, D. M. (2023). Mineral dust aerosol impacts on global climate and climate change. *Nature Reviews Earth & Environment*, *4*(2), 71-86.

Li, X., & Ginoux, P. (2025). An empirical parameterization to separate coarse and fine mode aerosol optical depth over land. *Geophysical Research Letters*, *52*(6), e2024GL114397.

Zender, C. S., Newman, D., & Torres, O. (2003). Spatial heterogeneity in aeolian erodibility: Uniform, topographic, geomorphic, and hydrologic hypotheses. *Journal of Geophysical Research: Atmospheres*, *108*(D17).

Zhao, A., Ryder, C. L., & Wilcox, L. J. (2022). How well do the CMIP6 models simulate dust aerosols?. *Atmospheric Chemistry and Physics*, *22*(3), 2095-2119.

Zomer, R. J., Xu, J., & Trabucco, A. (2022). Version 3 of the global aridity index and potential evapotranspiration database. *Scientific Data*, *9*(1), 409.

---

## Author Response (AR2)

We thank the reviewer for the additional, thoughtful comments on the revised manuscript. Our responses are provided below.

Thanks the authors for the clarification of their objective of this study and detailed responses. I have several comments to follow.

General Comments

1.      As the author responded in the general comments, the focus of their study is the interannual variability instead of absolute dust emission fluxes. I recommend modifying the title accordingly to add "interannual variability of windblown dust emissions" to avoid confusion.

Response: Thank you for the suggestion. The manuscript title has been revised to "The relative importance of wind and hydroclimate drivers in modulating the **interannual variability** of dust emissions in Earth system models".

2.      As a general comment for the focus of this study, if it is argued that there is no observational data available for the evaluation of dust emissions, the discussion of the interannual variability of dust emissions in this study is purely inter-model comparisons. Then the analyses are the simulated explainability of wind and hydroclimate parameters from different models, which could be the real relative importance or simply model failure. Then what would be the significance, or informative guidance from this study?

Response: A significant aspect of this study is comparing multi-model dust simulations through the lens of the relative importance of dust emission drivers, which reflects the inherent nature (i.e., unobservable, highly model-specific) of simulated dust emission fluxes. This presents a new framework for diagnosing model behaviors and biases. While our analysis is based on inter-model comparison, it reveals potential biases in how models represent the physical coupling between dust emissions and their driving factors. For instance, GFDL-ESM4 and CESM2-CAM-Kok are found to overestimate the hydroclimate influence and underestimate the wind influence over hyperarid regions.

3.      As the relative importance of wind and hydroclimate parameters is the focus of this study, and wind speed is especially sensitive to model resolution. I recommend recognize the importance of model resolution for careful interpretation of the relative importance of wind and hydroclimate parameters. Their importance could be swapped at different spatial resolutions.

Response: We agree that model resolution plays an important role in simulating near-surface wind strength and total dust emission fluxes. But, because most models tune the threshold wind velocity or soil erodibility maps to produce reasonable dust emission patterns, we expect the model resolution to have small effects on the *temporal covariability between wind speed and dust emission*s. This is supported by our results

over the hyperarid climate zone, where dust emission is predominantly controlled by wind speed in most ESMs, regardless of the model resolution. Models with finer resolutions (e.g., CESM and GFDL-ESM4) do not necessarily capture the dominant wind control in the hyperarid zone, compared to coarser-resolution models.

---

## Author Response (AR3)

Due to the recent U.S. government shutdown, Referee #1 was unable to submit their comments during the formal review period. They have now followed up with additional comments, I kindly request that you address them in a revised manuscript, so the published paper avoids potential misunderstandings with model developers and is strengthened for the community.

Response: Sorry about this unfortunate situation. We are very grateful for the editor's efforts in compiling the new comments and coordinating this new round of revision.

1. Incorporate or acknowledge available observational constraints used in dust-model intercomparisons
The reviewer reiterates that most dust–emission intercomparison studies incorporate some type of observational reference — whether satellite AOD, in-situ flux data, or campaign-based measurements (e.g. FRAGMENT field campaigns). They suggest the manuscript would benefit from acknowledging the existence of such datasets and briefly explaining why they were not used here. Since you previously explored MODIS AOD and found it unsuitable for your purpose, you may wish to expand the explanation and comment on the limitations or relevance of other observational datasets.

Response: During the first-round review, we used more than two decades' MODIS Level-2 deep blue aerosol records to compare with the ESMs and MERRA-2 reanalysis. We used the Li and Ginoux (2025) approach to isolate dust from total AOD and the Ginoux et al. (2012) approach to isolate regions frequently affected by persistent dust loadings. As seen in Fig. 1 below, the total $R^2$ or variance in MODIS dust AOD (or DAOD) explained by the six predictors is very low globally, compared to the $R^2$ values (>0.8) in MERRA-2 dust emissions (see Fig. 2 below). Our analysis shows that the MODIS DAOD record is not a good proxy for dust emission fluxes for temporal variability analysis, probably because DAOD is affected not only by emission, but also by atmospheric transport, dry/wet removal, and size partitioning processes.

Granted dust emission flux measurements are available from field campaigns, these measurements are from only a few locations and short time periods, making them unsuitable for comparison with multidecadal global model simulations considered in this study.

In the revised manuscript (Lines 92-97) we acknowledged the existence of satellite AOD and in situ data and explained why they are not used in this study:
"Although satellite-derived dust AOD and in-situ dust measurements provide valuable constraints on dust variability (e.g., Prospero and Lamb, 2003; Voss and Evan, 2020), they integrate the effects of emission, transport, and deposition, making it difficult to isolate the emission process itself. Also, due to lack of global validation data, we focus on diagnosing inter-model inconsistency in representing the dust emission variability

and its physical controls, rather than validating individual model performance against observations."

[Figure]

Figure 1. Total $R^2$ explained by MERRA-2 wind speed and hydroclimate drivers in the MODIS DAOD.

[Figure]

Figure 2. Total $R^2$ explained by MERRA-2 wind speed and hydroclimate drivers in the MERRA-2 dust emission fluxes. Copied from Fig. 7t in the manuscript.

2. Clarify/Remove the interpretation of GFDL-ESM4 dust-emission sensitivities
The reviewer emphasizes that several statements in the manuscript may mischaracterize dust-emission drivers in GFDL-ESM4. They point specifically to Section 2.5 of Shevliakova et al. (2024), which states that soil-moisture dependence is disabled in the CMIP6 version of GFDL-ESM4. This contradicts the interpretation in Line 470 that "anomalous sensitivities to soil

moisture" explain model behavior. They also stress that the GFDL scheme contains additional controlling factors beyond wind, soil moisture, and LAI, including both LAI (green vegetation) and SAI (brown vegetation), which can introduce low-frequency variability. For these reasons, the reviewer strongly recommends that all material related to GFDL-ESM4 be removed as doing so would be simpler than revising the manuscript in depth. If you decide not to fully remove the GFDL-ESM4 results, please provide sufficient justification. In my view, the bottom line is to remove any discussions and conclusions attributing the GFDL-ESM4 behavior to soil-moisture sensitivity.

Response: Thanks for providing the additional information about GFDL-ESM4. The dominance analysis is purely statistically driven, and independent from the physical linkages between dust emission and the considered drivers. In the case of GFDL-ESM4, since the soil moisture effect is disabled in the model, the strong hydroclimate influence from dominance analysis is very likely driven by spurious statistical covariations between the model-simulated dust emission fluxes and the topsoil water content. Indeed, Shevliakova et al. (2024) reported that GFDL-ESM4 overestimates the top-layer (0-5 cm) soil water content over global drylands compared with satellite estimates, with positive biases of up to 100% in regions such as central North Africa, South Africa, northwestern China, southwestern USA and Australia (see their Fig.9). This overestimated soil moisture may covary with the dust emission fluxes, even if soil moisture is not used in the dust parameterizations in GFDL-ESM4. To avoid misunderstandings and per the reviewer's request, we decided to remove GFDL-ESM4 from the paper. In the revision, we removed all content related to GFDL-ESM4, and added CanESM5.0, alongside CanESM5.1 which was already included.

In addition, we acknowledge that we did not consider all the physical drivers represented in the model, but only focus on six common physical drivers in order to ensure a fair comparison of all ESMs. This is now clarified in the revised paper as follows.

At Lines 171-174, we added:
"Note that we donot not include all the physical drivers represented in each model because of limited data availability in the CMIP6 online archive, and because some models incorporate additional drivers not used by others. Hence we focus on a common set of six readily available predictors to provide a consistent and fair comparison across the ESMs and MERRA-2 reanalysis."

At Lines 426-430 (Section 4 Conclusions), we added:
"Note that the physical drivers considered in this study may not fully represent all the dust emission driving factors for specific emission schemes; instead, we focus on a common set of drivers for all models to provide a fair comparison across the ESMs.

Therefore, the inferred relative importance from this analysis is limited to those common drivers considered and their influences on dust emissions in different models. Also, because of the statistical nature of dominance analysis, the predictor importance results shall be interpreted with caution when linking to model parameterizations."

3. Acknowledge other region-specific drivers
The reviewer notes that factors such as SAI, land-use change, and related vegetation dynamics play important roles in regions such as the Sahel, India, Australia, and the western United States. They recommend that the manuscript explicitly acknowledge these influences and clarify that the dominance analysis does not include all possible drivers in each Earth System Model.

Response: We acknowledge that our analysis does not consider all the physical drivers represented in every model, but focuses on six common physical drivers in order to provide a fair comparison of all ESMs. Also, not all the physical drivers (such as stem area index) are available in the CMIP6 online archive. This is now clarified in the revised paper as follows.

At Lines 171-174, we added:
"Note that we donot not include all the physical drivers represented in each model because of limited data availability in the CMIP6 online archive, and because some models incorporate additional drivers not used by others. Hence we focus on a common set of six readily available predictors to provide a consistent and fair comparison across the ESMs and MERRA-2 reanalysis."

At Lines 426-430 (Section 4 Conclusions), we added:
"Note that the physical drivers considered in this study may not fully represent all the dust emission driving factors for specific emission schemes; instead, we focus on a common set of drivers for all models to provide a fair comparison across the ESMs. Therefore, the inferred relative importance from this analysis is limited to those common drivers considered and their influences on dust emissions in different models. Also, because of the statistical nature of dominance analysis, the predictor importance results shall be interpreted with caution when linking to model parameterizations."

4. Concern about broader mischaracterization of multiple models
The reviewer mentions that GFDL-ESM4 is not the only ESM potentially mischaracterized in the manuscript. They fear this could create long-term confusion in the community about how dust emissions are represented in specific models. Because contacting ESM model developers at this stage may be too late, the reviewer urges careful removal or significant softening of model-specific conclusions unless supported directly by model documentation.
I appreciate your patience and understanding with this unusual late-review situation. Please feel free to reach out if you need clarification on any of the reviewer's points.

Response: We have removed all content related to GFDL-ESM4. We also carefully revised the paper by cross-referencing with original model documentations. In the revision, we also suggest caution in interpreting the statistically inferred predictor influences to model parameterizations.

At Lines 426-430 (Section 4 Conclusions), we added:
"Note that the physical drivers considered in this study may not fully represent all the dust emission driving factors for specific emission schemes; instead, we focus on a common set of drivers for all models to provide a fair comparison across the ESMs. Therefore, the inferred relative importance from this analysis is limited to those common drivers considered and their influences on dust emissions in different models. Also, because of the statistical nature of dominance analysis, the predictor importance results shall be interpreted with caution when linking to model parameterizations."

References

Ginoux, P., Prospero, J. M., Gill, T. E., Hsu, N. C., & Zhao, M. (2012). Global-scale attribution of anthropogenic and natural dust sources and their emission rates based on MODIS Deep Blue aerosol products. *Reviews of Geophysics*, *50*(3).

Li, X., & Ginoux, P. (2025). An empirical parameterization to separate coarse and fine mode aerosol optical depth over land. *Geophysical Research Letters*, *52*(6), e2024GL114397.